# Soil nitrous oxide emissions from global land ecosystems and their drivers within the LPJ-GUESS model (v4.1)

Jianyong Ma[1], Almut Arneth[1,2], Benjamin Smith[3,4], Peter Anthoni[1], Xu-Ri[5], Peter Eliasson[6], David Wårlind[4], Martin Wittenbrink[1], Stefan Olin[4]

[1]Institute of Meteorology and Climate Research-Atmospheric Environmental Research, Global Land-Ecosystem Modelling group, Karlsruhe Institute of Technology, Garmisch-Partenkirchen, Germany

[2]Institute of Geography and Geoecology, Karlsruhe Institute of Technology, Karlsruhe, Germany

[3]Hawkesbury Institute for the Environment, Western Sydney University, Richmond, NSW, Australia

[4]Department of Physical Geography and Ecosystems Science, Lund University, Lund, Sweden

[5]Key Laboratory of Alpine Ecology and Biodiversity, Institute of Tibetan Plateau Research, Chinese Academy of Sciences, Beijing, China

[6]Department of Earth and Environmental Sciences, Botswana International University of Science and Technology, Palapye, Botswana

*Correspondence to:*

Jianyong Ma (jianyong.ma@kit.edu) and Stefan Olin (stefan.olin@nateko.lu.se)

**Abstract**

Nitrogen (N) transformation processes by soil microbes account for significant nitrous oxide ($N_2O$) emissions from natural ecosystems and cropland. However, understanding and quantifying global soil $N_2O$ emissions and their responses to changing environmental conditions remain challenging. Here, we implemented a soil nitrification-denitrification module into the dynamic vegetation model LPJ-GUESS to estimate $N_2O$ emissions from global lands. The performance of this new development is examined using observed $N_2O$ fluxes from natural soil and cropland field trials, and independent global-scale estimates. LPJ-GUESS broadly reproduces the cumulative $N_2O$ emissions under different climate conditions and N fertilizer applications that are observed in the field experiments, with some deviations in emission seasonality. Globally, simulated soil $N_2O$ emissions from terrestrial ecosystems increase from 5.6±0.2 Tg N $yr^{-1}$ in the 1960s to 9.9±0.3 Tg N $yr^{-1}$ in the 2010s, with croplands contributing about two thirds of the total increase. East Asia and South Asia show the fastest growth rates in $N_2O$ emissions over the study period due to the expansion of fertilized croplands. On a global scale, N fertilization (including synthetic fertilizer and manure use), atmospheric N deposition, and climate change contribute 58%, 46%, and 24%, respectively, to the simulated soil $N_2O$ emissions in the 2010s. Rising $CO_2$ levels in the atmosphere reduce the simulated emissions by 32% through increased plant N uptake, whereas land-use changes have varied spatial effects on emissions depending on N management intensity after land-cover conversion. Our estimates only account for the direct soil $N_2O$ emissions, excluding those from fertilized pasture. This study highlights the importance of environmental factors in influencing global soil $N_2O$ emissions, particularly for assessing greenhouse gas mitigation potential in agricultural ecosystems.

**1 Introduction**

Understanding how anthropogenic activities that influence greenhouse gas emissions and exchanges affect the climate system is crucial to address the threats posed by global change. While much of the attention has been on the carbon (C) cycle and carbon dioxide ($CO_2$), interests in non-$CO_2$ greenhouse gases have grown over the last two decades, particularly for nitrous oxide ($N_2O$), due to its rapidly increased concentration in the atmosphere (from ~290 ppb in 1940 to ~336 ppb in 2022, with a marked rapid growth since 1980 of 1.3 ppb $yr^{-1}$ in 2022; Tian et al., 2024). This human-induced $N_2O$ increase significantly contributes to global warming, as $N_2O$ is about 273 times more potent than $CO_2$ at warming the atmosphere in a 100-year perspective (Canadell et al., 2021). Additionally, $N_2O$ is an important stratospheric ozone-depleting substance, potentially increasing surface levels of harmful ultraviolet radiation (World Meteorological Organization, 2022).

The main sources of terrestrial $N_2O$ emissions to the atmosphere are nitrogen (N) transformation processes by soil microbes. Other major non-soil contributors include fossil fuel combustion, inland waters, and biomass burning (Tian et al., 2020). From 2000–2016, the global soil release of $N_2O$ through nitrification and denitrification from land ecosystems has been estimated to be 9–13 Tg N yr$^{-1}$ (Bouwman et al., 2013; Tian et al., 2019; Zaehle, 2013), with tropical rainforest soils being the most important sources due to their high organic matter turnover rates in the warm and moist environment (Stehfest and Bouwman, 2006). Agricultural soils (within and outside the tropics) are also critical owing to high reactive N levels. A significant portion of the increase in atmospheric $N_2O$ from agriculture can be attributed to N management practices and N applications to croplands, especially mineral fertilizer from the Haber-Bosch process. Estimates indicate that global cropland $N_2O$ emissions have increased by 1–3 times in the last several decades, ranging from 0.4–1.4 Tg N yr$^{-1}$ in the 1960s to 1.3–3.3 Tg N yr$^{-1}$ in the 2010s (Tian et al., 2019; Wang et al., 2020; Xu et al., 2020). These soil $N_2O$ emissions are expected to continue to rise due to the growing food demand associated with the increasing human population and changes in per-capita consumption (Davidson and Kanter, 2014; Fowler et al., 2013). Significant reductions of agricultural $N_2O$ emissions are required to achieve ambitious climate targets, in particular through changes in land management practices and, ultimately, enhanced N use efficiency—that is, increasing the fraction of N input that is harvested as products (Gu et al., 2023; Springmann et al., 2018; Zhang et al., 2021).

N input, especially the use of industrial N fertilizer, is one of the most reliable predictors to quantify past and present $N_2O$ emissions on agricultural ecosystems, with an estimated emission factor (EF) varying from 0.2–1.8% of the N applied to the soil (default value in IPCC Tier 1; Hergoualc'h et al., 2019). While large-scale estimates of $N_2O$ emissions have been made based on regional and/or global mean EF values (primarily estimated from various reactive N sources; see Crutzen et al., 2008; Smith et al., 2012), all EF methods assume a linear increase of $N_2O$ in response to N input. However, field experimental evidence shows that emission trends are usually exponential when N fertilizer rates exceed plant needs (Shcherbak et al., 2014; Song et al., 2018). Process-based ecological models that capture soil-vegetation interactions and effects of environmental drivers on physiological and biogeochemical processes of plants/crops, soils and microbial communities provide an approach for quantifying historical and future $N_2O$ emissions on large spatial scales (Butterbach-Bahl et al., 2013). This is due to their mechanistic parametrization of C-N dynamics between vegetation and soils under changing environmental conditions and land-use management (Pongratz et al., 2018).

Compared with some well-developed site-specific models in the 1990s (e.g., DAYCENT, Parton et al., 1996; DNDC, Li et al., 1992), the implementation of $N_2O$-related processes into global biosphere models, such as Dynamic Global Vegetation Models (DGVMs; Cramer et al., 2001) only began in the early years of this century. For instance, Xu-Ri and Prentice (2008) adopted the descriptions of nitrification and denitrification processes from the DNDC model (Li et al., 2000) and introduced them in a simplified way into the DGVM LPJ. They assumed that $N_2O$ fluxes are mainly regulated by soil moisture and temperature, carbon supply, soil characteristics (such as aeration and texture), and reactive N availability. Likewise, Zaehle et al. (2011) largely followed the approach of Xu-Ri and Prentice (2008) and incorporated inorganic soil N dynamics into the O-CN vegetation model, enhancing it with inclusion of soil pH and chemo-nitrification processes. To better represent the influence of soil microbes on N transformation, some DGVMs further considered the activity of nitrifiers and denitrifiers by simulating the growth and mortality of the responsible bacteria (e.g., CLM3.5, Saikawa et al., 2013; IBIS, Ma et al., 2022b; TRIPLEX, Zhang et al., 2017b). However, most of these DGVM developments focused predominantly on $N_2O$ emissions on natural soils, given the extensive coverage of natural vegetation on Earth's land surface (e.g., forest and grassland; see Huang and Gerber, 2015; Saikawa et al., 2013; Xu-Ri et al., 2012). Considering the growing contribution of the agricultural sector to the global N cycle, DGVMs are also being equipped to account for cropland management options—such as N fertilizer and manure use (Tian et al., 2012; Von Bloh et al., 2018), lime and basalt application (Val Martin et al., 2023), tillage (Ciais et al., 2011; Lutz et al., 2020), cover crops (Ma et al., 2023; Olin et al., 2015b; Porwollik et al., 2022), and residue retention (Ren et al., 2020). These strategies have been shown through field experiments to play crucial roles in regulating $N_2O$ emissions (Abalos et al., 2022; Li et al., 2023; Quemada et al., 2020; Yangjin et al., 2021).

In this study, we implemented soil nitrification-denitrification processes into the DGVM LPJ-GUESS, where detailed agricultural management practices had been incorporated previously (Olin et al., 2015a; Smith et al., 2014). Building on the concepts introduced by Xu-Ri and Prentice (2008), these processes are parameterized within our model to estimate soil $N_2O$ emissions across major natural vegetation and cropland land cover classes, allowing us to investigate how $N_2O$ responds to environmental changes and N management. The performance of the updated model is evaluated using field experimental data, published results from other modelling studies, and global inversion estimates. Our objective is to quantify the temporal and spatial pattern of global $N_2O$ fluxes across natural vegetation, pastures, and croplands, while exploring the environmental factors that drive changes in $N_2O$ emissions over the historical period.

## 2 Methods and data

### 2.1 Model description

The LPJ-GUESS model is a comprehensive, process-based global vegetation simulator designed to study plant-soil interactions and their influence on ecosystem biogeochemical cycling, including C-N dynamics of natural and managed ecosystems under environmental changes (e.g., global warming and $CO_2$ rising; see Smith et al., 2014) and land management (capturing e.g., conservation agriculture, forest thinning and clear-cutting; see Lindeskog et al., 2021; Olin et al., 2015b). This is achieved by simulating physiological and biogeochemical processes of plants on a daily basis (Smith et al., 2014). In the default global configuration, the model's natural vegetation component includes 12 plant functional types (PFTs): 10 forest and two grass types, each characterized by distinct phenological and morphological traits, bioclimatic constraints, and specific strategies for establishment, growth, and mortality. Pasture ecosystems are modelled as a competition between $C_3$ and $C_4$ grasses, with 50% of the aboveground biomass removed annually to represent grazing effects. To account for internal manure deposition from livestock on pastures, the model assumes that 75% of the N from the harvested biomass is returned to the soil (Lindeskog et al., 2013). Cropland in the version of LPJ-GUESS used in this study is characterized by six crop functional types (CFTs): two temperate $C_3$ cereals sown in spring and autumn/winter, a $C_4$ crop representing maize, a tropical $C_3$ crop representing rice, and two N-fixing $C_3$ grain legumes representing soybean and pulses (Ma et al., 2022a; Olin et al., 2015a). These CFTs are simulated as either rain-fed or irrigated, determined by the prescribed fractions provided as external input. Crops are harvested once every year when the required heat units are reached. Agricultural practices—such as tillage intensity, N mineral fertilizer and manure application, crop residue removal, and leguminous and non-leguminous cover crops—are also included (Ma et al., 2023; Olin et al., 2015b; Pugh et al., 2015). For large-scale application, to reflect the widespread adoption of conventional practices on current global agriculture (Porwollik et al., 2019), the model assumes that all croplands are under tillage management without cover cropping systems, and that 25% of aboveground crop residue is retained in the fields after harvest. Industrial N fertilizer is added to soils at three different stages of crop growth, with application rates varying by CFT. In contrast, all manure is applied as a single input at crop sowing to reflect the time required for manure N to become available for plant uptake in real-world practices (Olin et al., 2015b).

In the model, the C transfers induced by decomposition between 11 soil organic matter (SOM) pools drive N mineralization-immobilization processes, in order to maintain prescribed C:N ratios and mass balance in both the C-donor and C-receiver pools (Smith et al., 2014). Soil mineral N after the processes of

mineralization and immobilization (i.e., $NH_4^+$), along with biological N fixation ($NH_4^+$), chemical N-fertilizer input ($NO_3^-$ and $NH_4^+$), and atmospheric N deposition ($NO_3^-$ and $NH_4^+$), jointly determine the total size of the soil reactive N pool. This pool is initially depleted by vegetation uptake, followed by hydrological N losses and gaseous N emissions from the soils (Wårlind et al., 2014). Mineral N leaching in the model increases with soil $NO_3^-$ concentration linearly and is dynamically adjusted by soil water holding capacity and percolation rate. Gaseous N losses through soil N transformation are implemented in this study and described in detail below.

## 2.2 Representation of gaseous N emissions from the soil

### 2.2.1 Ammonia (NH₃) volatilization

N losses through $NH_3$ volatilization significantly affect the concentration of $NH_4^+$ in soils, thereby influencing N₂O emissions through nitrification. Following Xu-Ri and Prentice (2008), production of $NH_3$ from $NH_4^+$ is simulated as a response function to (1) the exchangeable $NH_4^+$ in the soil solution, (2) soil temperature, and (3) pH level:

$$\begin{cases} NH_{3soil} = NH_{4soil\_solution}^+ \cdot f_{t\_NH_3} \cdot f_{pH\_NH_3} \\ NH_{4soil\_solution}^+ = NH_{4soil}^+ \cdot W_{soil\_relative} \end{cases} \tag{1}$$

where $NH_{3soil}$ is the $NH_3$ concentration in the soil (kg N m⁻² d⁻¹); $NH_{4soil\_solution}^+$ is the exchangeable $NH_4^+$ concentration determined by the pool size of $NH_{4soil}^+$ and the relative soil water content ($W_{soil\_relative}$); $f_{t\_NH_3}$ and $f_{pH\_NH_3}$ are the limitations (ranging from 0 to 1) on volatilization by soil temperature and pH, respectively, and are taken from the LPJ-DyN (Xu-Ri and Prentice, 2008) and CLM5.0 (Val Martin et al., 2023) models:

$$f_{t\_NH_3} = \min[1, e^{308.56 \cdot (\frac{1}{71.02} - \frac{1}{T_{soil}+46.02})}] \tag{2}$$

$$f_{pH\_NH_3} = \begin{cases} 0.6 & \forall pH \in [0,5) \\ 0.6 + \frac{0.4}{3} \cdot (pH - 5) & \forall pH \in [5,8) \\ 1.0 & \forall pH \in [8, \infty) \end{cases} \tag{3}$$

where $T_{soil}$ is the soil temperature(℃) at a depth of 25 cm representing the mean temperature of the 0–50 cm top layer in the 2-layer implementation of LPJ-GUESS we used; pH is the soil pH values provided together with soil mineral properties as external input.

After $NH_3$ volatilization, the remaining soil $NH_4^+$ pool will continue to be depleted by nitrification processes in the model (see Fig. 1). Since nitrification and denitrification can occur simultaneously under

aerobic and anaerobic conditions, respectively, the concept of 'anaerobic balloon' proposed by Li et al. (2000) is adopted to partition reactive N species (e.g., $NO_3^-$ and $NH_4^+$) into these two soil states, with soil water-filled pore space (WFPS) used as an indicator (Xu-Ri and Prentice, 2008). According to Li et al. (2000), the size of the 'anaerobic balloon' increases exponentially with soil moisture. Therefore, in the model the fraction of anaerobic substrates ($f_{anaero}$) is simply estimated as an exponential function of WFPS:

$$\begin{cases} f_{anaero} = 0.05 + \dfrac{0.95 - 0.05}{1 + e^{-7.5 \cdot (WFPS - 0.5)}} \\ WFPS = \dfrac{W_{soil\_volume}}{POR_{soil}} \end{cases} \quad (4)$$

where $W_{soil\_volume}$ is the soil volumetric water content in the top layer (m³/m³; 0–50 cm); $POR_{soil}$ is the soil porosity determined by soil physical properties (m³/m³). The partitioning of soil substrates under aerobic ($X_{soil\_aero}$) and anaerobic ($X_{soil\_anaero}$) conditions is estimated using:

$$\begin{cases} X_{soil\_aero} = (1 - f_{anaero}) \cdot X_{soil} \\ X_{soil\_anaero} = f_{anaero} \cdot X_{soil} \end{cases} \quad (5)$$

where $X_{soil}$ is the soil substrate concentration (kg N (or C) m$^{-2}$ d$^{-1}$), representing any of the following in this study: $NH_4^+$, $NO_3^-$, $NO_2^-$, or labile carbon.

**2.2.2 Nitrification**

Autotrophic nitrification and heterotrophic nitrification are two distinct biological processes involved in the N transformation in soil ecosystems. We focused solely on representing autotrophic nitrification, which is the dominant process in most natural and agricultural soils (Chapin III et al., 2011). The heterotrophic pathway is also more challenging to model as it requires estimation of dissolved organic nitrogen as the main substrate for the responsible nitrifying bacteria. Autotrophic nitrification is an aerobic process wherein $NH_4^+$ undergoes sequential oxidation to $NO_2^-$ and then to $NO_3^-$, producing $NO_x$ and $N_2O$ as intermediates and/or by-products. The initial oxidation step involves two distinct groups of nitrifiers: ammonia-oxidizing bacteria and archaea. The subsequent oxidation of $NO_2^-$ to $NO_3^-$ is facilitated by nitrite-oxidizing bacteria. Due to the current limitation in the model's ability to simulate the growth and mortality of soil microbes, we integrate these two oxidation steps into one single process—i.e., $NH_4^+$ is oxidized to $NO_3^-$ directly—to collectively represent nitrification in LPJ-GUESS (see Fig.1). The production of $NO_3^-$ through nitrification ($R_{NO_3^-\_nit}$) is formulated as:

$$R_{NO_3^-\_nit} = k_{max\_nit} \cdot NH_{4soil\_aero}^+ \cdot f_{t\_nit} \cdot f_{WFPS\_nit} \cdot f_{pH\_nit} \quad (6)$$

where $k_{max\_nit}$ is the maximum nitrification coefficient and set as a constant of 0.1 based on the experimental data from Khalil et al. (2004); $NH_{4soil\_aero}^{+}$ is the aerobic soil $NH_4^{+}$ concentration after $NH_3$ volatilization (kg N m$^{-2}$ d$^{-1}$; see Sect. 2.2.1); $f_{t\_nit}$, $f_{WFPS\_nit}$, and $f_{pH\_nit}$ are the limitation factors by soil temperature, moisture, and pH, respectively. Soil temperature plays a crucial role in regulating microbial activities. For nitrite-oxidizing bacteria, 37–39°C is found to be optimal for substrate oxidation (Taylor et al., 2019) and for ammonia-oxidizing bacteria and archaea the optimal soil temperature can range from 31–42°C (Ouyang et al., 2017). In the model, the maximum nitrification rate is thus assumed to occur at 38°C, as the average optimal temperature for these three groups of nitrifiers:

$$f_{t\_nit} = (\frac{70 - T_{soil}}{70 - 38})^{12} \cdot e^{12 \cdot \frac{T_{soil} - 38}{70 - 38}} \tag{7}$$

Besides soil temperature, soil moisture and pH are also key factors affecting nitrification rates. Gleeson et al. (2010) demonstrated that the activity of nitrifying bacteria decreases rapidly when soil WFPS exceeds 0.6, and stops completely when it surpasses 0.8. Consequently, a three-threshold limitation function is incorporated into LPJ-GUESS to simulate soil moisture influence (Eq. 8). For soil pH constraints, the response function of nitrification rate is adopted from Parton et al. (2001), as implemented in the DAYCENT model (Eq. 9):

$$f_{WFPS\_nit} = \begin{cases} 0.239 \cdot e^{2.38 \cdot WFPS} & \forall WFPS \in [0.0, 0.6] \\ 1 - \frac{1}{0.2} \cdot (WFPS - 0.6) & \forall WFPS \in (0.6, 0.8) \\ 0 & \forall WFPS \in [0.8, 1.0] \end{cases} \tag{8}$$

$$f_{pH\_nit} = 0.56 + \frac{arctan[0.45 \cdot \pi \cdot (pH - 5)]}{\pi} \tag{9}$$

Since the mechanisms of N-gas emissions during nitrification are not yet fully understood (Butterbach-Bahl et al., 2013), we adopted the same assumption as other ecosystem and crop models, that the gaseous N concentration through nitrification (i.e., $R_{gas\_nit}$) is proportional to nitrification rate, and estimate the emissions using:

$$R_{gas\_nit} = F_{max\_gas\_nit} \cdot R_{NO_3^-\_nit} \tag{10}$$

where $R_{NO_3^-\_nit}$ is the soil $NO_3^-$ concentration produced through nitrification in Eq. 6 (kg N m$^{-2}$ d$^{-1}$); $F_{max\_gas\_nit}$ is the maximum fraction of nitrified N lost as $NO_x$ and $N_2O$, a parameter that varies widely between models due to differences in their structure configuration for simulating N-gas species and specific nitrification processes (Gabbrielli et al., 2024). We assume this parameter as a constant of 0.25, based on the MicN model (Ma et al., 2022b), to broadly account for the potential N-gas fluxes driven by

different groups of nitrifying bacteria. Previous studies have revealed that during nitrification, $NO_x$ dominates at a WFPS below 0.3, and the ratio of $NO_x:N_2O$ is often close to 1.0 at a WFPS of 0.6 (Davidson et al., 2000; Pilegaard, 2013). Thus, in the model, nitrified gaseous N concentration (i.e., $R_{gas\_nit}$) is partitioned into $NO_x$ and $N_2O$ species using an empirical function of soil WFPS:

$$\begin{cases} NO_{x\_nit} = P_{NO_x:(NO_x+N_2O)\_nit} \cdot R_{gas\_nit} \\ N_2O_{\_nit} = (1 - P_{NO_x:(NO_x+N_2O)\_nit}) \cdot R_{gas\_nit} \\ P_{NO_x:(NO_x+N_2O)\_nit} = 1 - \dfrac{0.5}{1 + e^{-20 \cdot (WFPS-0.375)}} \end{cases} \quad (11)$$

where $NO_{x\_nit}$ and $N_2O_{\_nit}$ are nitrified $NO_x$ and $N_2O$ gases in the soil, respectively (kg N m$^{-2}$ d$^{-1}$); $P_{NO_x:(NO_x+N_2O)\_nit}$ is the partitioning scheme between $NO_x$ and $N_2O$ taken from previous studies (Goldberg and Gebauer, 2009; Pilegaard, 2013).

### 2.2.3 Denitrification

Denitrification is a series of reduction reactions driven by different groups of microorganisms in anaerobic conditions. Heterotrophic denitrifying bacteria facilitate the full reduction chain from $NO_3^-$ to molecular nitrogen ($N_2$) ($NO_3^- \rightarrow NO_2^- \rightarrow NO_x \rightarrow N_2O \rightarrow N_2$), a process known as denitrifier denitrification. In contrast, autotrophic nitrifiers typically convert $NH_4^+$ to $NO_2^-$ under aerobic conditions, but when oxygen becomes scarce, they switch to reducing $NO_2^-$ to $NO_x$ and $N_2O$, and finally to $N_2$, a process known as nitrifier denitrification. Given the high reactivity of $NO_x$ under the reducing conditions that facilitate denitrification (Parton et al., 2001; Schlüter et al., 2024), accurately simulating every single transformation step from $NO_2^-$ to $N_2$ is challenging due to the interdependent nature of these processes (Ma et al., 2022b). In this study, following the concept of 'holes-in-a-pipe' (e.g., Firestone and Davidson, 1989; Davidson et al., 2000; Val Martin et al., 2023), we combine the entire reduction chain from $NO_2^-$ to $N_2$ into a single step (see Fig. 1) to broadly represent the emissions of all N gases produced during denitrification:

$$DENIT_{LPJ-GUESS} = NO_{3soil\_anaero}^- \rightarrow R_{NO_2^-\_denit} \rightarrow R_{gas\_denit} \quad (12)$$

where $DENIT_{LPJ-GUESS}$ is the N reduction chain during denitrification represented in LPJ-GUESS; $NO_{3soil\_anaero}^-$ is the soil $NO_3^-$ concentration under anaerobic conditions calculated in Eq. 5 (kg N m$^{-2}$ d$^{-1}$); $R_{NO_2^-\_denit}$ and $R_{gas\_denit}$ are the denitrified soil $NO_2^-$ and all N-gas concentration (kg N m$^{-2}$ d$^{-1}$), respectively. Since the availability of the particular N oxide ($NO_3^-$ or $NO_2^-$) and soil labile C are the two dominant drivers controlling the activity of denitrifiers (Weier et al., 1993; Chapin III et al., 2011), we incorporate these two

factors into the model to reflect their limitations on the denitrification rate, along with soil temperature as an additional constraint:

$$R_{NO_2^-\_denit} = k_{max\_denit} \cdot NO_{3soil\_anaero}^- \cdot f_{t\_denit} \cdot f_{LC\_denit} \cdot f_{NO_3^-\_denit} \tag{13}$$

where $f_{t\_denit}$, $f_{LC\_denit}$ and $f_{NO_3^-\_denit}$ are limitations (ranging from 0 to 1) on denitrification by soil temperature, available C, and soil $NO_3^-$ concentration, respectively. $k_{max\_denit}$ is the maximum denitrification coefficient and can reach 1.0 when there are no limitations by environmental factors or the populations of denitrifying bacteria (Gabbrielli et al., 2024). In LPJ-GUESS, to reflect the absence of limitation due to the growth of denitrifiers, $k_{max\_denit}$ is assumed to be a constant value of 0.5, chosen as the middle point of the possible range from 0 to 1. For $f_{LC\_denit}$ and $f_{NO_3^-\_denit}$, both response functions are adopted from the DNDC model (Li et al., 1992), following the Michaelis-Menten equation. For $f_{t\_denit}$, an empirical sigmoid function, built on experimental observations (Benoit et al., 2015; Ma et al., 2022b), is used for the parametrization of temperature effects:

$$\begin{cases} f_{LC\_denit} = \dfrac{Rh_{soil\_anaero}}{K_C \cdot W_{soil\_volume} + Rh_{soil\_anaero}} \\[3mm] f_{NO_3^-\_denit} = \dfrac{NO_{3soil\_anaero}^-}{K_N \cdot W_{soil\_volume} + NO_{3soil\_anaero}^-} \\[3mm] f_{t\_denit} = e^{(-1) \cdot \frac{(T_{soil} - 37)^2}{25^2}} \end{cases} \tag{14}$$

where $K_C$ and $K_N$ are Michaelis-Menten constants of 0.017 kg C m$^{-3}$ and 0.083 kg N cm$^{-3}$ for labile C and N oxides, respectively (Li et al., 1992). $W_{soil\_volume}$ is volumetric soil water content in the top layer (m$^3$/m$^3$; 0–50 cm). Since labile C is not explicitly modelled in LPJ-GUESS, we use soil heterotrophic respiration under anaerobic conditions ($Rh_{soil\_anaero}$; kg C m$^{-2}$ d$^{-1}$) as a surrogate for C availability, following Parton et al. (2001) and Xu-Ri and Prentice (2008).

In addition to soil labile C and N oxides, much experimental evidence has shown that soil pH and moisture are also critical in regulating the denitrification rate, particularly during the transformation process from $NO_2^-$ to $N_2$ (e.g., Bao et al., 2012; Bergaust et al., 2010; Kool et al., 2011; Lim et al., 2018). Therefore, in the model, after the first reduction step ($NO_{3soil\_anaero}^- \rightarrow R_{NO_2^-\_denit}$), soil $NO_2^-$ is further denitrified as N gases ($R_{gas\_denit}$) using the reduction equation as the first step, but with added limitations by soil pH and WFPS (Eqs. 15-17). Both of these two response functions ($f_{pH\_NO_2^-\_denit}$ and $f_{WFPS\_NO_2^-\_denit}$ below) are established based on experimental data, and are taken from Blanc-Betes et al. (2021) and Ma et al. (2022b), respectively:

$$R_{gas\_denit} = k_{max\_denit} \cdot R_{NO_2^-\_denit} \cdot f_{t\_denit} \cdot f_{LC\_denit} \cdot f_{NO_2^-\_denit} \cdot f_{pH\_NO_2^-\_denit} \cdot f_{WFPS\_NO_2^-\_denit} \quad (15)$$

$$f_{pH\_NO_2^-\_denit} = \begin{cases} 0.001 & \forall pH \in [0,4] \\ \frac{pH-4}{3} & \forall pH \in (4,7) \\ 1.0 & \forall pH \in [7,\infty) \end{cases} \quad (16)$$

$$f_{WFPS\_NO_2^-\_denit} = 0.624 + \frac{0.8 \cdot \arctan[0.45 \cdot \pi \cdot (10 \cdot WFPS - 8)]}{2.85} \quad (17)$$

Davidson et al. (2000) and Pilegaard (2013) pointed out that $NO_x$ emissions exponentially decrease when soil WFPS exceeds 0.3 and cease entirely at 0.7. We therefore assume that the denitrified gaseous N ($R_{gas\_denit}$ in Eq. 18) is to produce $NO_x$ and $N_2O$ species only when WFPS is below 0.7. Above this threshold, the production shifts to $N_2O$ and $N_2$ gases instead.

$$R_{gas\_denit} = \begin{cases} NO_{x\_denit} + N_2O_{\_denit} & \forall WFPS \in [0.0, 0.7) \\ N_2O_{\_denit} + N_2{\_denit} & \forall WFPS \in [0.7, 1.0] \end{cases} \quad (18)$$

where $NO_{x\_denit}$, $N_2O_{\_denit}$, and $N_2{\_denit}$ are denitrified $NO_x$, $N_2O$, and $N_2$ in the soil, respectively (kg N m$^{-2}$ d$^{-1}$). For soil WFPS between 0 and 0.7, the denitrified gaseous N is partitioned into $NO_x$ and $N_2O$ using:

$$\begin{cases} N_2O_{\_denit} = P_{N_2O:(NO_x+N_2O)\_denit} \cdot R_{gas\_denit} \\ NO_{x\_denit} = (1 - P_{N_2O:(NO_x+N_2O)\_denit}) \cdot R_{gas\_denit} \\ P_{N_2O:(NO_x+N_2O)\_denit} = \max[0, (3.2 \cdot WFPS - 0.92)/(3.2 \cdot WFPS - 0.08)] \end{cases} \quad (19)$$

where $P_{N_2O:(NO_x+N_2O)\_denit}$ is the partitioning ratio between $NO_x$ and $N_2O$, built on the data provided in Davidson et al. (2000) and Pilegaard (2013). Previous studies indicated that low temperature, combined with low pH and low soil water content, reduce the activity of $N_2O$ reductase, thereby increasing the ratio of $N_2O$ to ($N_2 + N_2O$) in the last step of denitrification (Weier et al., 1993; Siljanen et al., 2020). Accordingly, for soil WFPS above 0.7, the partitioning of $R_{gas\_denit}$ to $N_2O$ and $N_2$ is determined by soil temperature, moisture, and pH levels, with emissions estimated using:

$$\begin{cases} N_2O_{\_denit} = P_{N_2O:(N_2+N_2O)\_denit} \cdot R_{gas\_denit} \\ N_{2\_denit} = (1 - P_{N_2O:(N_2+N_2O)\_denit}) \cdot R_{gas\_denit} \\ P_{N_2O:(N_2+N_2O)\_denit} = f_{N_2O:(N_2+N_2O)\_denit\_t} \cdot f_{N_2O:(N_2+N_2O)\_denit\_WFPS} \cdot f_{N_2O:(N_2+N_2O)\_denit\_pH} \end{cases} \quad (20)$$

$$\begin{cases} f_{N_2O:(N_2+N_2O)\_denit\_t} = \dfrac{1}{1 + e^{(T_{soil}-5)/10}} \\ f_{N_2O:(N_2+N_2O)\_denit\_WFPS} = 0.85 - \dfrac{0.85 - 0.10}{1 + e^{-23 \cdot (WFPS - 0.75)}} \\ f_{N_2O:(N_2+N_2O)\_denit\_pH} = \min(1, 7.23 \cdot \dfrac{1}{e^{0.497 \cdot pH}}) \end{cases} \quad (21)$$

where $P_{N_2O:(N_2+N_2O)\_denit}$ is the partitioning ratio of $N_2O$ to $(N_2 + N_2O)$, jointly constrained by soil temperature ($f_{N_2O:(N_2+N_2O)\_denit\_t}$), WFPS ($f_{N_2O:(N_2+N_2O)\_denit\_WFPS}$), and pH value ($f_{N_2O:(N_2+N_2O)\_denit\_pH}$). The field-based observed $N_2O:(N_2 + N_2O)$—data from Weier et al. (1993) for soil temperature; Maag and Vinther (1996) for soil WFPS; Liu et al. (2010) and Rochester (2003) for soil pH—are used to establish the three limitation functions.

### 2.2.4 Gas diffusion

In the model, N gases transformed through ammonia volatilization, nitrification, and denitrification accumulate in the soil and are ultimately released into the atmosphere. Since soil aeration status and temperature are key factors influencing gas diffusion (Li et al., 1992; Zhang et al., 2017b), a straightforward equation based on these two variables, as suggested by Xu-Ri and Prentice (2008), is adopted to estimate gas transport from the top soil layer to the atmosphere:

$$f_{t\_dif} = \min[1, e^{308.56 \cdot (\frac{1}{68} - \frac{1}{T_{soil}+46})}] \tag{22}$$

$$\begin{cases} NH_{3\_gas} = NH_{3soil} \cdot f_{t\_dif} \cdot (1 - WFPS) \\ NO_{x\_gas} = (NO_{x\_nit} + NO_{x\_denit}) \cdot f_{t\_dif} \cdot (1 - WFPS) \\ N_2O_{\_gas} = (N_2O_{\_nit} + N_2O_{\_denit}) \cdot f_{t\_dif} \cdot (1 - WFPS) \\ N_{2\_gas} = N_{2\_denit} \cdot f_{t\_dif} \cdot (1 - WFPS) \end{cases} \tag{23}$$

where $f_{t\_dif}$ is soil temperature limitation function on gas diffusion and taken from Xu-Ri & Prentice (2008); $NH_{3\_gas}$, $NO_{x\_gas}$, $N_2O_{\_gas}$, and $N_{2\_gas}$ are $NH_3$, $NO_x$, $N_2O$, and $N_2$ gases released from the soil to the atmosphere, respectively. It should be noted that other N-gas processes, such as gas diffusion fluxes between soil layers and $NO_x$ losses due to its rapid oxidation near the plant canopy, are not implemented in the model at the moment. A flowchart of key transformations of soil mineral N in LPJ-GUESS is shown in Fig. 1.

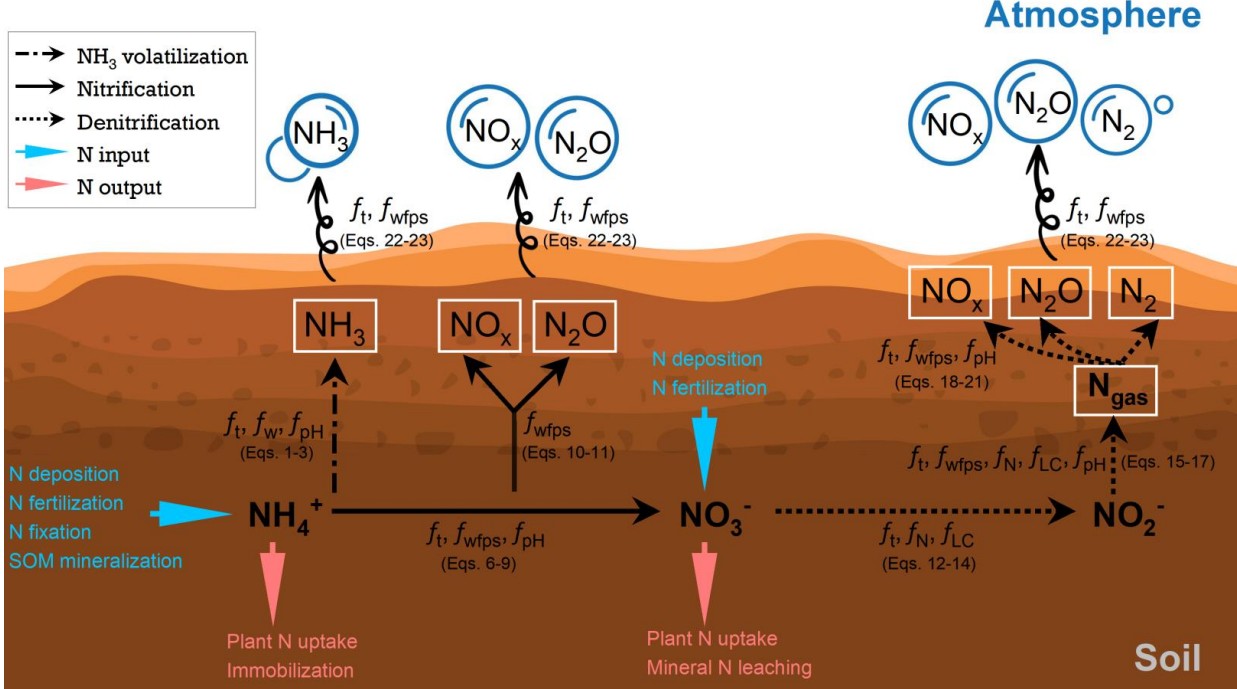

**Figure 1.** Representation of key transformations of soil mineral N in LPJ-GUESS. N species in white boxes represent the concentration of the transformed gases accumulated in the soil. See the Equations in Sections 2.2.1–2.2.4 for details on the limitation factors during each process.

## 2.3 Model experimental protocol

This study includes three model applications or experiments. First, we evaluate the model performance in reproducing observed $N_2O$ fluxes from various natural and cropland field sites worldwide. Next, a global simulation is performed over the historical period to assess the model results against estimates from other modelling studies and inversion approaches. In the last part, we analyze and discuss the environmental factors driving temporal and spatial changes in historical $N_2O$ emissions on global natural vegetation, pasture and cropland ecosystems.

To initialize the model state, we adopted the spin-up procedure following the protocol in our earlier study (Ma et al., 2022a). All simulations are initialized with a 500-year spin-up using atmospheric $CO_2$ concentration and N deposition from 1901, along with repeating de-trended 1901–1920 climate (data sources described below). The spin-up starts with simulations of potential natural vegetation, representing unmanaged forest and grassland ecosystems. In the last 30 years of the spin-up, the model gradually introduces cropland by linearly increasing the cropland fraction from zero to the 1901 historic value. Details of the model experiment protocol are provided below.

**2.3.1 Model evaluation at site scale**

To assess model performance, field-based $N_2O$ observations from natural and cropland soils were
compiled from the published literature. This dataset includes 49 natural vegetation and 55 cropland sites
situated between ~43°S and ~61°N (Figs. 2a-2b). For natural lands, field studies typically reported annual
$N_2O$ emissions (kg N ha$^{-1}$ yr$^{-1}$) across five ecosystems: tropical forests (13 sites), temperate forests (21
sites), boreal forests (5 sites), grasslands (8 sites), and savannas (2 sites). These measurements were
recorded during the period from 1981 to 2010. In contrast, observed data from croplands focused solely
on cumulative $N_2O$ emissions over the crop growing season within four major cropping systems (kg N ha$^{-1}$
season$^{-1}$; wheat, maize, rice, and legumes). This dataset, covering the period from 1994 to 2020,
examined how $N_2O$ fluxes respond to varying N-fertilizer inputs and management practices on agricultural
soils (Fig. 2c). Detailed information for these sites—including geographic coordinates, experimental
periods, and cropping management systems—is provided in Tables S1-S2 in the Supporting Information.

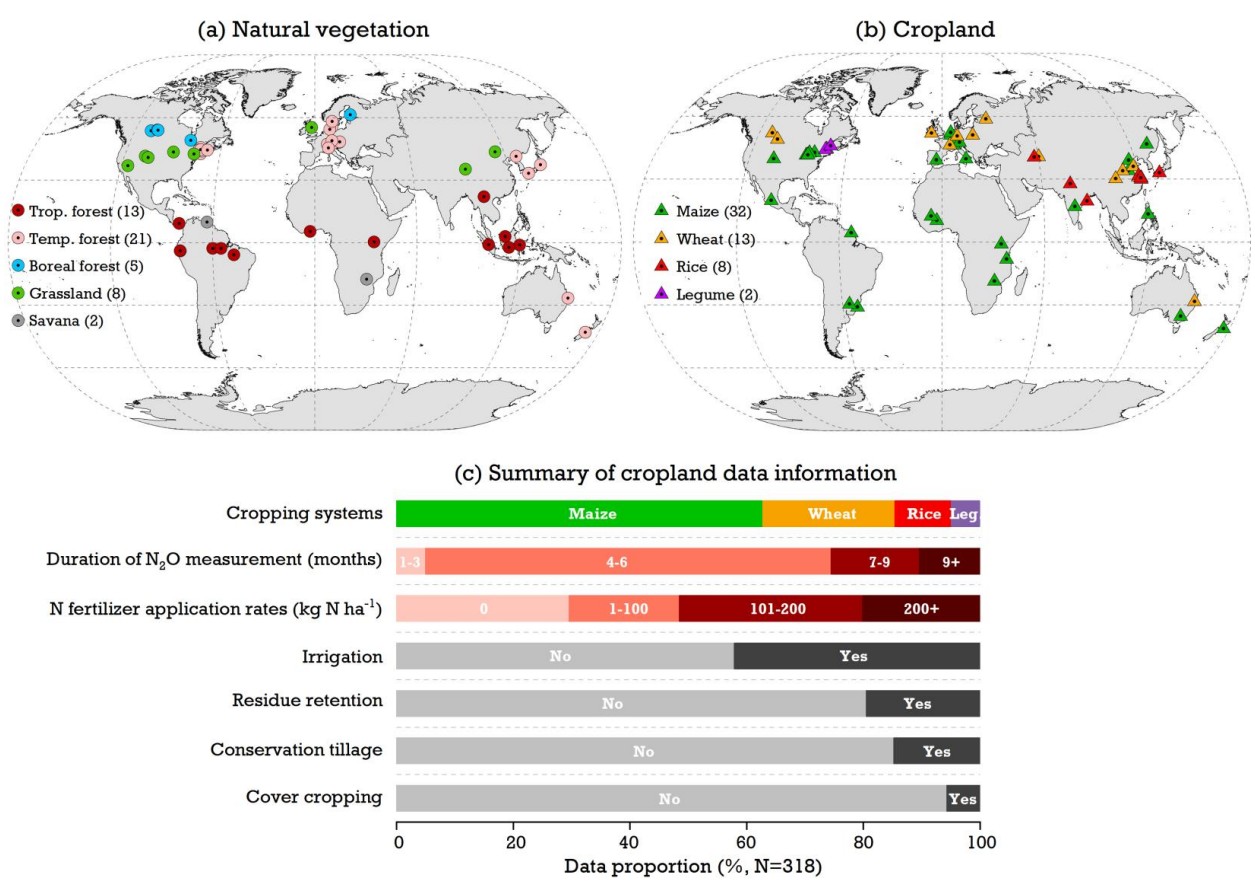

**Figure 2.** Distribution of in-situ $N_2O$ observation studies used for model evaluation on (a) natural vegetation and (b)
cropland. In cropland field experiments, overall $N_2O$ fluxes measurement information—cropping systems,
measurement duration, and N fertilizer and conservation agriculture managements (i.e., residue, tillage, and cover
crops)—was shown in (c).

Due to the lack of weather and N deposition data for most study sites, an observation-based gridded climate dataset, CRUJRA v2.4 (Harris et al., 2020; Kobayashi et al., 2015), and an atmospheric N deposition dataset simulated by CCMI ($NH_x$-N and $NO_y$-N; Tian et al., 2018), were used as inputs to drive LPJ-GUESS, selecting the representative grid cell (0.5°×0.5°) for each experimental site. To maintain equilibrium in soil C and N pools after model spin-up, natural vegetation and cropland systems were simulated continuously

from 1901 onwards at their respective experimental sites. Since N-fertilizer management had been established for several years before the start of the $N_2O$ emission trials at most cropland sites, we assumed a five-year period of N fertilizer use in croplands prior to the field trials, with application rates consistent with those used during the trials. Throughout the experimental period, simulations on croplands were performed based on the management details reported in the literature (Table S2 in

Supporting Information), whereas natural vegetation sites remained unmanaged (i.e., growing under rain-fed and unfertilized conditions without timber or biomass harvests). Additionally, to estimate soil hydraulic properties and evaluate our developed N transformation processes, soil physical characteristics—such as soil pH and texture (i.e., content of sand, silt, and clay)—were collected from the literature and kept constant during the simulation period. The accuracy of the simulated $N_2O$ fluxes was

statistically assessed using adjusted $R^2$ (the goodness of fit for the linear regression), mean error (ME), mean absolute error (MAE), and the root mean square error (RMSE) across all sites.

**2.3.2 Global soil $N_2O$ emissions and their drivers**

For global-scale applications, climate variables—daily temperature (minimum, mean, and maximum), precipitation, solar radiation, specific humidity, and wind speed from CRUJRA v2.4 dataset—were used for

driving model simulations, ranging from 1901–2020 at a resolution of 0.5°×0.5° (Harris et al., 2020; Kobayashi et al., 2015). Historical annual atmospheric $CO_2$ concentration and monthly N deposition rates over the same period were derived from Meinshausen et al. (2020) and Tian et al. (2018), respectively (Fig. S1 in Supporting Information). Land use and land cover data spanning from 1901 to 2020 were sourced from HILDA+ (Winkler et al., 2021), initially at a 0.01° resolution and later aggregated to 0.5°. This dataset

provides annually varying proportions of natural vegetation, pasture, and cropland for each grid cell. The crop distribution map, including rain-fed and irrigated fractions per grid cell around the year 2000, was extracted from the MIRCA dataset (Portmann et al., 2010) and aggregated to match the six CFTs simulated in the model (see Sect. 2.1 above). To parameterize soil water characteristics, global gridded soil profile data at 0.5° resolution from WISE3 (Batjes, 2009) were used. Synthetic N fertilizer and manure application

rates to crops were obtained from Ag-GRID (Elliott et al., 2015) and Zhang et al. (2017a), respectively,

covering the period of 1901–2015 (Fig. S1 in Supporting Information). As N fertilizer data were available only until 2015, we assumed that N application rates to croplands during 2016–2020 remained steady at 2015 levels. At present, pasture ecosystems represented in LPJ-GUESS do not receive any N fertilizer inputs.

All model experiments for this part of the study spanned from 1901 to 2020. However, the focus of our analyses was primarily on the period from 1960–2020, during which N fertilizer use became prevalent. In the 'Reference' simulation (referred to as S0 in Table 1), the model was driven by a constant recycled 1901–1920 climate, together with 1901 $CO_2$ concentration, N deposition, land use, and fertilizer inputs, to monitor model drift and internal variation. The result of this run was used to generate background $N_2O$

emissions with minimal human influence. Conversely, the 'All_Combined' run (referred to as S1 in Table 1) incorporated dynamic inputs for all these factors from 1901–2020, reflecting realistic $N_2O$ emissions due to anthropogenic perturbation and environmental change. In each subsequent simulation (referred to as S2–S6 in Table 1), all but one of the factors were allowed to vary dynamically over time, with one factor held constant at its 1901 level. This setup was designed to isolate and identify the individual impact of

each factor on $N_2O$ emissions by comparing S1 with any runs in S2–S6 (denoted as $\Delta N_2O_{si}$ in Eq. 24). The total change caused by these five factors was calculated as the difference between S1 and S0 simulations ($\Delta N_2O_{all}$). The relative contribution (%) of every single factor to the total change in $N_2O$ emissions was then determined by the ratio of $\Delta N_2O_{si}$ to $\Delta N_2O_{all}$ using Eq. 24:

$$
\begin{cases}
\Delta N_2O_{si} = N_2O_{s1} - N_2O_{si} & i \in [2,6] \\
\Delta N_2O_{all} = N_2O_{s1} - N_2O_{s0} & \\
\Delta N_2O_{si\%} = \Delta N_2O_{si} / \Delta N_2O_{all} \times 100\% & i \in [2,6] \\
\Delta N_2O_{interactive\%} = \left(\Delta N_2O_{all} - \sum \Delta N_2O_{si}\right) / \Delta N_2O_{all} \times 100\% & i \in [2,6]
\end{cases}
\tag{24}
$$

where $\Delta N_2O_{si\%}$ represents the relative contribution of each factor to the total change in $N_2O$ emissions,

and $\Delta N_2O_{interactive\%}$ denotes the interactive effects between these factors, presented as a percentage. Si refers to the five environmental factor simulations of Const_Climate, Const_$CO_2$, Const_Ndep, Const_Nfert, and Const_LUC given in Table 1.

**Table 1.** Simulation setups representing the contribution of environmental factors to global soil $N_2O$ emissions (see Section 2.3.2).

| Sim. Code | Sim. Name | Climate | $CO_2$ | N deposition | N fertilization[b] | LUC |
|---|---|---|---|---|---|---|
| S0 | Reference | 1901–1920[a] | 1901 | 1901 | 1901 | 1901 |
| S1 | All_Combined | 1901–2020 | 1901–2020 | 1901–2020 | 1901–2020[c] | 1901–2020 |
| S2 | Const_Climate | 1901–1920[a] | 1901–2020 | 1901–2020 | 1901–2020[c] | 1901–2020 |
| S3 | Const_$CO_2$ | 1901–2020 | 1901 | 1901–2020 | 1901–2020[c] | 1901–2020 |
| S4 | Const_Ndep | 1901–2020 | 1901–2020 | 1901 | 1901–2020[c] | 1901–2020 |
| S5 | Const_Nfert | 1901–2020 | 1901–2020 | 1901–2020 | 1901 | 1901–2020 |
| S6 | Const_LUC | 1901–2020 | 1901–2020 | 1901–2020 | 1901–2020[c] | 1901 |

a–Historical climate (1901–1920) with temperature de-trended, repeated throughout the period 1901–2020; b–N fertilization on croplands, including mineral N fertilizer and manure application. Fertilized pasture is not simulated in this study; c–Historical N inputs between 1901–2015, with the 2015 data extended to cover the period 2016–2020.

## 3 Results

### 3.1 Model-observation comparisons at site scale

#### 3.1.1 Model performance across all sites

The simulated cumulative $N_2O$ emissions generally showed a good agreement with measurements, with regression slopes ranging from 0.72 to 0.87. However, the model tended to overestimate measured emissions by 65% globally for natural vegetation and 11% for cropland sites (Figs. 3a–3b). According to in-situ observations on natural lands (Fig. S2 in Supporting Information), tropical forests were identified as the primary $N_2O$-emitting sources, showing a mean cumulative flux of 1.23 kg N ha$^{-1}$ annually. Temperate forests followed with an average flux of 0.52 kg N ha$^{-1}$ yr$^{-1}$, and boreal forests had the lowest emissions at 0.12 kg N ha$^{-1}$ yr$^{-1}$. The model broadly reproduced a similar regional pattern with the highest $N_2O$ emissions in the warm tropics and the lowest emissions in the cold boreal region, although it underrepresented $N_2O$ sources in the tropics by 15% while overestimating emissions in temperate and boreal forests by 84% and 50%, respectively. Both field measurements and model experiments indicated

grasslands to be a weak $N_2O$ source, with the simulations being 24% lower than the observed emissions of 0.19 kg N ha$^{-1}$ yr$^{-1}$ on average. This pattern of underestimation was also found at the grass-dominant savanna site, where LPJ-GUESS underestimated the reported $N_2O$ fluxes of the field trials by 78% (Fig. S2 in Supporting Information).

Compared with natural vegetation, cropping systems showed higher $N_2O$ emissions throughout the growing period, primarily due to their high levels of N fertilizer use (Fig. 3c). While the linear regression slopes for the four simulated cropping systems were not far from 1.0, LPJ-GUESS overestimated the seasonal $N_2O$ fluxes in most cropland measurements. This overestimation was particularly pronounced in field-grown rice trials, wherein the modelled emissions were approximately 50% higher than the field observations. Conversely, grain legumes—crops that fix atmospheric N and typically receive little N fertilizer in the fields—were the only cropping system showing lower simulated $N_2O$ emissions than measured, with an underestimation of 45% (Fig. 3c). A positive exponential relationship between N fertilizer input and cumulative $N_2O$ emissions was found across various field trials in observations. Although LPJ-GUESS simulations effectively captured this observed increase in $N_2O$ fluxes with rising N fertilizer levels, some discrepancies were noted. Specifically, the model underestimated emissions in the unfertilized soils by 28% and overestimated emissions in most highly fertilized trials (>200 kg N ha$^{-1}$) by 65%. This overestimation tended to become more pronounced with higher N application rates (Fig. S2 in Supporting Information).

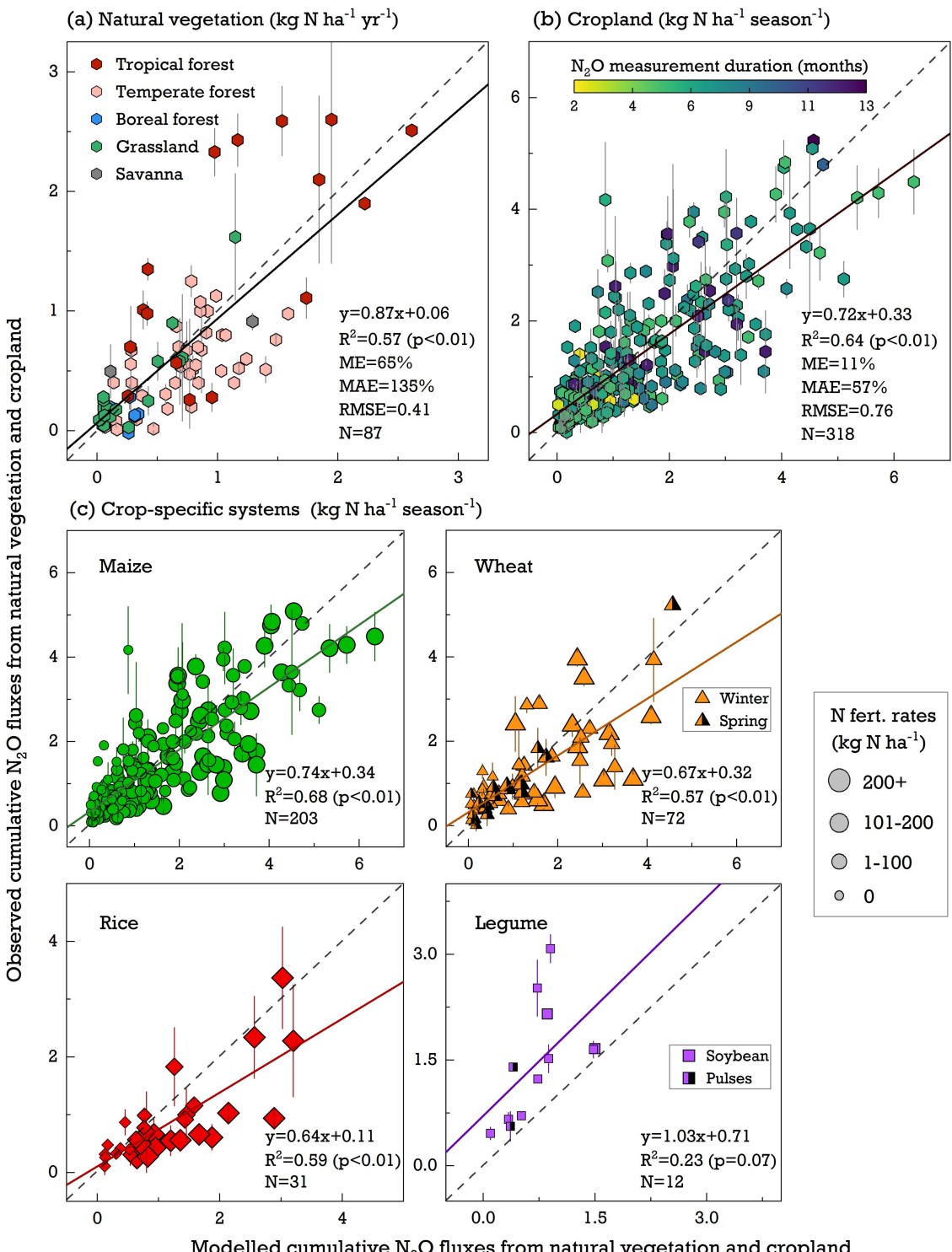

**Figure 3.** Comparison of modelled and observed cumulative N₂O emissions from (a) natural vegetation, (b) cropland, and (c) crop-specific systems across all studied sites. The dashed line is the 1:1 line and the black bold line is a fitted linear regression. ME (mean error) and MAE (mean absolute error) are shown as percentages, while RMSE (root mean square error) is in kg N ha$^{-1}$ yr$^{-1}$ for natural vegetation and in kg N ha$^{-1}$ season$^{-1}$ for cropland. Error bars denote the standard error from different field trial replicates collected from literature. In (c), marker size from large to small indicates descending N fertilizer rates applied to crops.

### 3.1.2 Seasonal pattern of N$_2$O emissions on natural vegetation

The seasonal pattern of N$_2$O emissions showed significant variation across different vegetation ecosystems and between individual years in both field experiments and model simulations. For a tropical rainforest in Brazil (Fig. 4a), observed fluxes of N$_2$O increased during the rainy season and rapidly decreased over the dry period. The model was able to reproduce this rainfall-induced mean response; however, the simulated peak flux was delayed by two months compared with the observations, likely due to a temporal mismatch between the modelled and reported soil moisture during the wet season (Fig. S3 in Supporting Information; Davidson et al., 2008). In contrast, a tropical montane forest in Indonesia (Fig. 4b), where rainfall is more evenly distributed throughout the year (Purbopuspito et al., 2006), exhibited no distinct seasonal pattern in N$_2$O fluxes in the simulations or the observations. Soil temperature and water availability jointly influenced the magnitude of N$_2$O emissions in temperate and boreal ecosystems, with the largest fluxes observed during summer (June–August) and weaker sources (or occasional sinks, depending on sites) during the winter season (Figs. 4c–4g). LPJ-GUESS did not capture these negative fluxes recorded in the field trials and instead produced near zero N$_2$O emissions in cold and dry conditions (Figs. 4f–4g), mainly as a result of negative temperatures inactivating nitrification and denitrification during the winter. Soil moisture was identified as the dominant factor controlling the seasonal dynamics of N$_2$O fluxes at a semi-arid grassland with a sandy loam soil texture, where observed WFPS ranged from 0.01–0.48 between June and August (Du et al., 2006). While the model effectively represented this N$_2$O rise due to increasing WFPS under aerobic conditions, it overestimated total emissions by ~95% over the summer season (Fig. 4h). This overestimation primarily resulted from the model simulating a higher WFPS value of 0.37 for this sand-dominant soil, compared with the observed average of 0.20 in these three months (Fig.S3 in Supporting Information; Du et al., 2006).

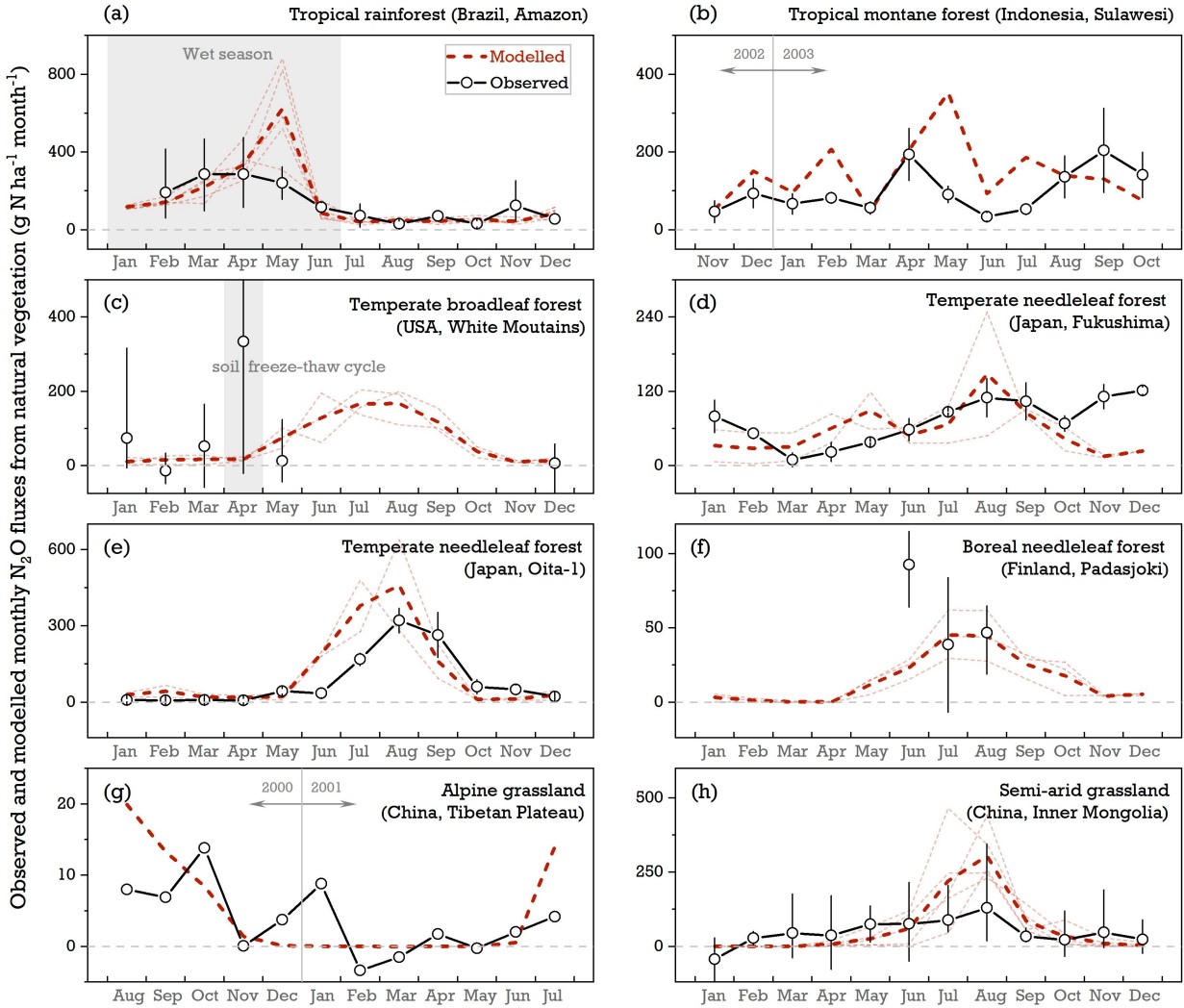

**Figure 4.** Comparison of modelled and observed monthly N$_2$O emissions at eight natural vegetation sites: (a) Tapajós National Forest, Brazil (54.9°W, 2.9°S; 2000–2004; Davidson et al., 2008); (b) Tropical montane forest, Central Sulawesi, Indonesia (120.3°E, 1.4°S; November 2002–October 2003 in Wuasa; Purbopuspito et al., 2006); (c) White Mountain National Forest, USA (71.8°W, 43.9°N; 1998–2000; Groffman et al., 2006); (d) Temperate cedrus forest, Fukushima, Japan (140.3°E, 37.4°N; 2003–2004; Morishita et al., 2007); (e) Temperate cedrus forest, Oita-1 site, Japan (131.3°E, 33.2°N; 2003–2004; Morishita et al., 2007); (f) Boreal spruce forest, Finland (24.9°E, 61.3°N; June–August of 2000, 2001, and 2003; Maljanen et al., 2006); (g) Alpine grassland, Tibetan Plateau, China (93.1°E, 35.1°N; August 2000–July 2001; Pei et al., 2004); and (h) Native semi-arid grassland, Inner Mongolia, China (114.7°E, 43.5°N; 1995, 1998, and 2001–2003; Du et al., 2006). The dark red dashed lines denote the multi-year average of simulations over the observation period. Dashed lines in lighter colors represent the simulations for individual years. Open circles indicate the observed N$_2$O fluxes averaged over all measurement years, with error bars showing the maximum and minimum values.

### 3.1.3 Cropland $N_2O$ emission response to N fertilization

The model's ability to simulate the observed $N_2O$ flux response to N fertilizer application was assessed using seasonal data from a rain-fed maize field site in Northeast China (133.5°E, 47.6°N). Alongside three levels of N fertilizer inputs (0, 150, and 250 kg N ha$^{-1}$, denoted as N0, N150, and N250 below), this cropping system was managed with conventional tillage, zero residue retention, and no cover crops (see Song and Zhang, 2009). Over the maize growing season, the cumulative $N_2O$ fluxes were measured at 0.4, 2.0, and 4.8 kg N ha$^{-1}$ for N0, N150, and N250 treatments, respectively, exceeding the simulated results of 0.2, 1.8, and 4.1 kg N ha$^{-1}$. The model's estimates of the fertilizer-induced $N_2O$ emission factors were 1.1% for the N150 treatment and 1.6% for the N250 treatment, which closely matched the measured range of 1.0–1.8%, suggesting a good overall agreement between the model simulations and field experiments regarding the $N_2O$ response to N addition (Figs. 5a–5c). However, LPJ-GUESS failed to capture the peak $N_2O$ fluxes at the maize flowering stage, particularly in the highly fertilized N250 treatment (Fig. 5c). The remaining difference between modelled and measured seasonal dynamics was found within the three-week period after each application of N fertilizer, with the simulated $N_2O$ rates being much higher than the observed values. This overestimation indicated that the soil N transformation processes in the model were overly sensitive to reactive N input. At this site with silt-clay textured soil, the simulated seasonal trends of soil WFPS and temperature broadly aligned with the observed variations (Pearson correlation coefficients of 0.42–0.57, $p < 0.05$ for both variables), despite the modelled WFPS values being consistently higher than the observed ones (Figs. 5d–5e).

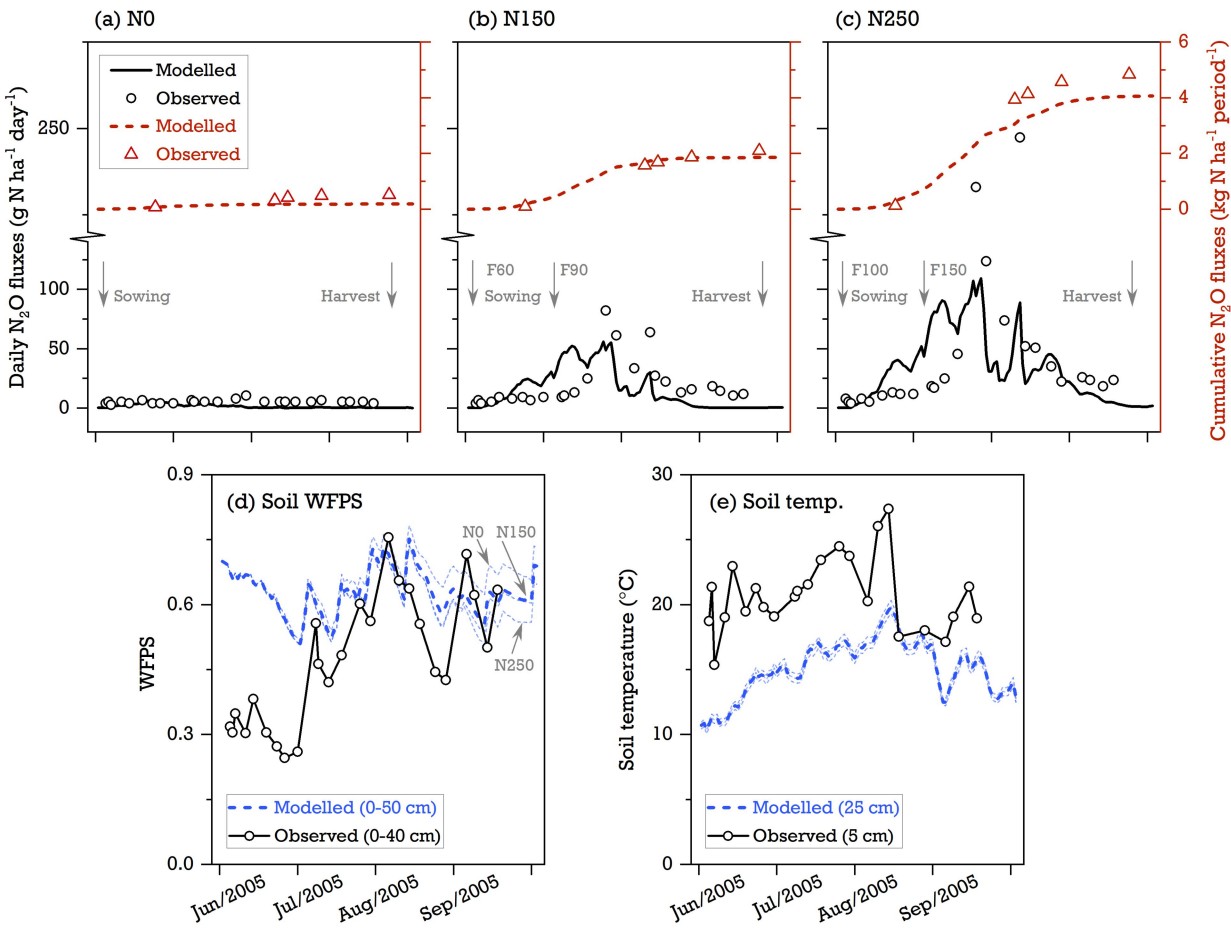

**Figure 5.** Modelled and observed seasonal pattern of (a-c) N₂O emissions in response to three levels of N fertilizer inputs (0, 150, and 250 kg N ha$^{-1}$, referred to as N0, N150, and N250); (d) soil moisture (WFPS); and (e) soil temperature in °C at a rain-fed maize field site in China for the cropping season 2005 (133.5°E, 47.6°N; Song and Zhang, 2009). In (a-c), a total of 40% and 60% of the mineral N fertilizer are applied at the time of maize sowing and jointing stage, respectively, with the implemented managements of conventional tillage, zero residue retention, and no cover crops (Song and Zhang, 2009). The thick blue dashed lines in (d-e) denote the simulated mean of three N fertilizer inputs. The thinner dashed lines represent the simulations for individual N treatments.

## 3.2 Global soil N₂O emissions

The modelled global N₂O emissions from the soil to the atmosphere increased steadily from 1960–2020, with estimates growing from 5.6±0.2 Tg N yr$^{-1}$ in the 1960s to 9.9±0.3 Tg N yr$^{-1}$ by the 2010s (Fig. 6). While natural soils remained the major sources of N₂O, their contributions to global total emissions declined from 81% to 59% over this period. In contrast, simulated N₂O emissions on croplands showed a clear upward trend since 1960, coinciding with the widespread use of synthetic N fertilizer. Croplands reached their highest average emission rate, 3.6±0.2 Tg N yr$^{-1}$, in the 2010's decade, representing 37% of global land emissions. Pasture ecosystems were identified as weak sources of N₂O in our simulations, with

historical estimates varying between 0.3±0.04 Tg N yr⁻¹ to 0.4±0.05 Tg N yr⁻¹ (Table S3 in Supporting Information), noting however that fertilized pastures were not simualted in the model. Overall, the model's estimates on global soil N₂O emissions since the 1980s—both in magnitude and interannual variability—were broadly consistent with other studies using bottom-up approaches (Global Nitrous Oxide Budget; Tian et al., 2024), process-based modelling (Global N₂O Model Inter-comparison Project, NMIP; Tian et al., 2019), and atmospheric inversion methods (Thompson et al., 2019).

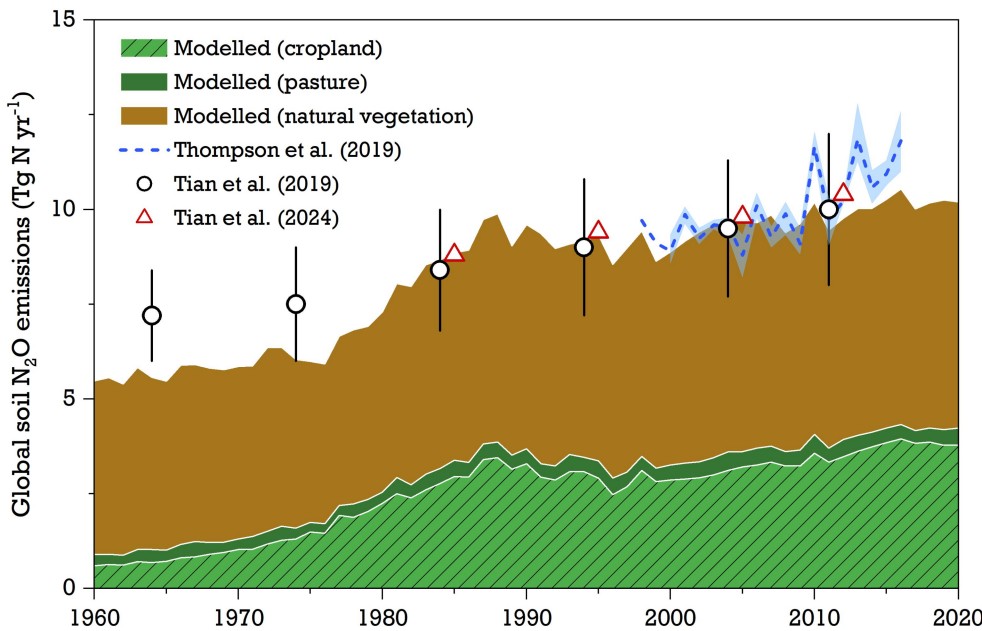

**Figure 6.** Modelled global soil N₂O emissions from natural vegetation, pasture, and cropland by LPJ-GUESS for the period 1960–2020 (see S1 run in Table 1), compared with global land estimates from the literature. Reported data from Tian et al. (2019) in open circles indicate 10-year average emissions simulated by seven process-based vegetation models, with error bars representing one standard deviation. The decade mean emissions from Tian et al. (2024) in red triangles are derived from bottom-up estimates and exclude the emissions from non-soil components (such as inland water, fossil fuels and industry, and biomass burning). Annual N₂O emissions between 1998–2016 from Thompson et al. (2019) in dashed line represent the average of three atmospheric inversion frameworks, with the range indicated by the blue shaded area.

The modelled map of soil N₂O emissions revealed large spatial variation in the 1960s (Fig. 7a). Simulated N₂O rates as high as 1.5–2.5 kg N ha⁻¹ yr⁻¹ were found in tropics (such as the Congo Basin) and parts of Europe and the United States, where neither water nor temperature was a critical constraint for nitrification and denitrification processes. Conversely, regions with arid climates or at high latitudes experienced N₂O emissions as low as 0–0.5 kg N ha⁻¹ yr⁻¹, as soil water content or temperature limitations restricted the turnover rates of soil N pools in LPJ-GUESS. At a regional scale, Africa and South America,

with their extensive areas of natural vegetation, together accounted for 50% of simulated global land $N_2O$ emissions in the 1960s. North America and Southeast Asia followed, with contributions of 13% and 9%, respectively (Table S3 in Supporting Information).

Compared with the 1960s, soil $N_2O$ emission rates increased in most parts of the world during 2011–2020, mainly as a result of the combined effects of environmental changes and N management practices (Fig. 7b). The regions with high N deposition and intensive fertilizer use—such as northern China, India, central Europe, and eastern United States (Fig. S1 in Supporting Information)—were simulated to have the highest $N_2O$ rates, ranging from 3.5 to 4.5 kg N $ha^{-1}$ $yr^{-1}$. Compared with other regions, East Asia and South Asia showed the fastest growth in emissions between the 1960s and 2010s (Figs. 7c–7l), largely due to their expansion of fertilized croplands. From 2011 to 2020, these two regions jointly contributed 31% to global total emissions, which was slightly higher than the combined 27% contribution from North America and Europe (Table S3 in Supporting Information).

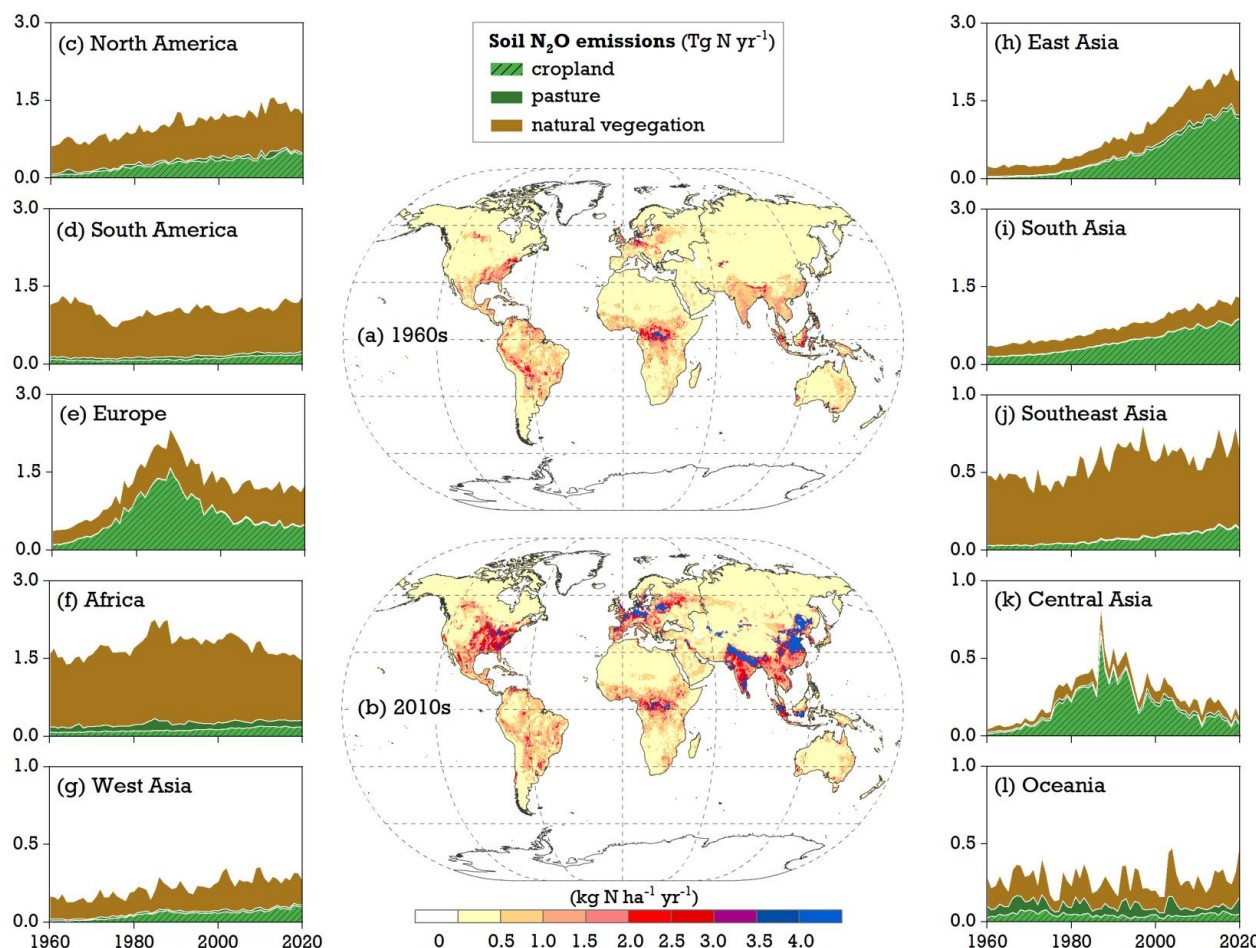

**Figure 7.** Maps of soil $N_2O$ emissions modelled by LPJ-GUESS in the decades of the (a) 1960s and (b) 2010s, and (c-l) time series of simulated total soil $N_2O$ emissions from natural vegetation, pasture, and cropland at a continental

level for the period 1960–2020 (see S1 run in Table 1). The division of the 10 continents used in this study is given in Fig. S4 in Supporting Information.

**3.3 Drivers of increased $N_2O$ emissions**

Changes in the temporal-spatial patterns of soil $N_2O$ emissions were influenced by a combination of land-use factors, climate variation, and atmospheric composition changes. Between 1960–2020, the global increase in soil $N_2O$ emissions was primarily driven by the growing use of N fertilizer and manure, as well as N deposition and climate change, which elevated soil levels of reactive N available for $N_2O$ production (Fig. 8). During the 2010s, N fertilization alone contributed 3.2±0.2 Tg N yr$^{-1}$, representing 58% of the increased global terrestrial emissions. N deposition and climate change followed, with estimated contributions of 46% and 24%, respectively (Table S4 in Supporting Information). In contrast, rising $CO_2$ concentrations were found to lower soil $N_2O$ emissions, with the negative effect increasing over time. This reduction was particularly significant in natural vegetation and pasture ecosystems and less pronounced in croplands. From 2011–2020, the $CO_2$ effect was simulated to reduce global soil $N_2O$ emissions by 1.83±0.1 Tg N yr$^{-1}$, roughly offsetting half of the increased emissions due to N fertilizer use.

In model simulations, the impact of land-use change on soil $N_2O$ emissions showed significant spatial variation depending on N management intensity after land-cover conversion (Fig. 9). For instance, increased $N_2O$ due to land-use change were typically found in regions where soils received high reactive N input (such as northern China and central Europe). Conversely, regions with low levels of N fertilizer and manure use—like most countries in Africa and South America—were beneficial in reducing $N_2O$ emissions after the conversion from natural vegetation to croplands. Additionally, a net expansion of natural lands in the northern temperate regions, such as the southeast U.S. and eastern Europe, was found to contribute to the mitigation of emissions. On a global scale, land-use-induced reduction in $N_2O$ emissions on natural lands was estimated at -0.60±0.05 Tg N yr$^{-1}$ during 2011–2020, while cropland experienced an increase of 0.58±0.04 Tg N yr$^{-1}$ over the same period.

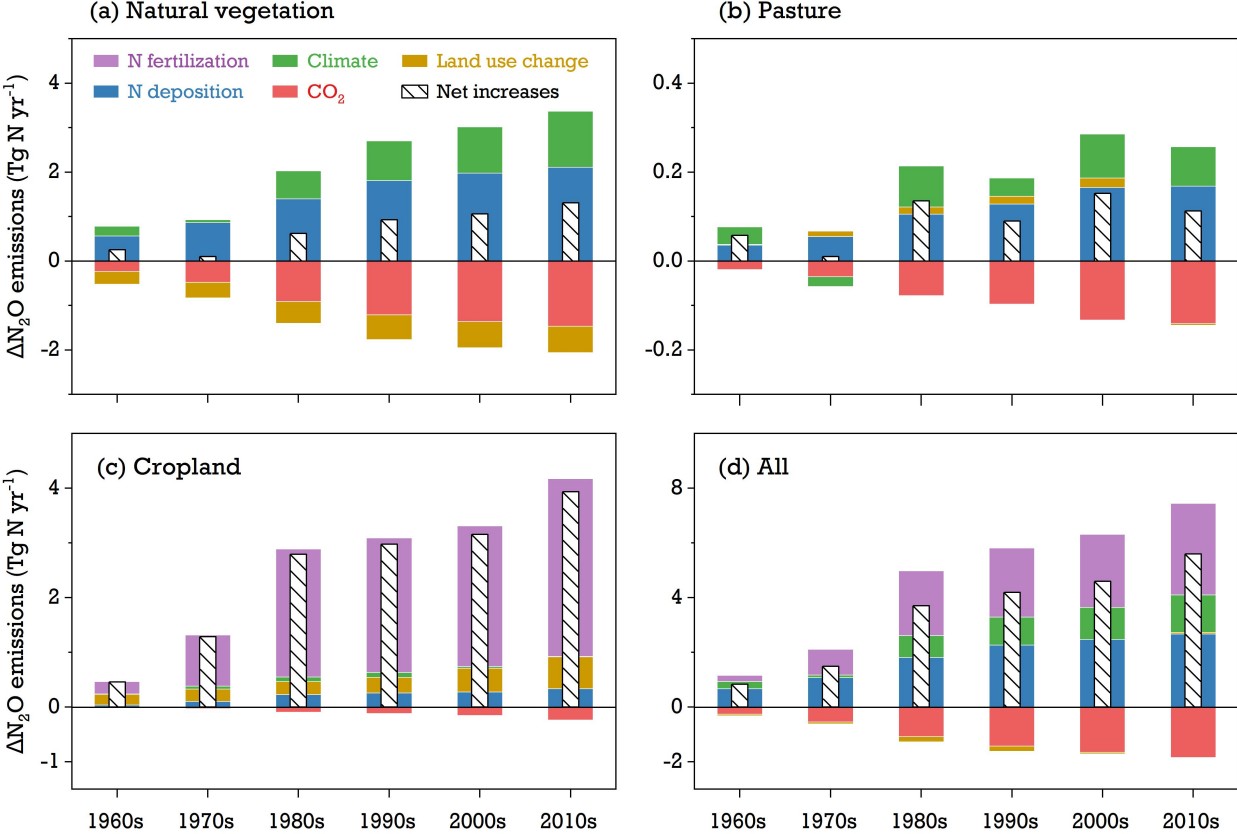

**Figure 8.** Modelled contributions ($\Delta N_2O$) of environmental factors (climate change, rising $CO_2$ levels, N fertilization, N deposition, and land use change) to global soil $N_2O$ emissions between 1960–2020 across various vegetation types: (a) natural vegetation, (b) pasture, (c) cropland, and (d) the aggregate of all three ecosystems. The white bar with slashes is the net emission from all factors' contribution. See Eq. 24 for $\Delta N_2O$ calculation.

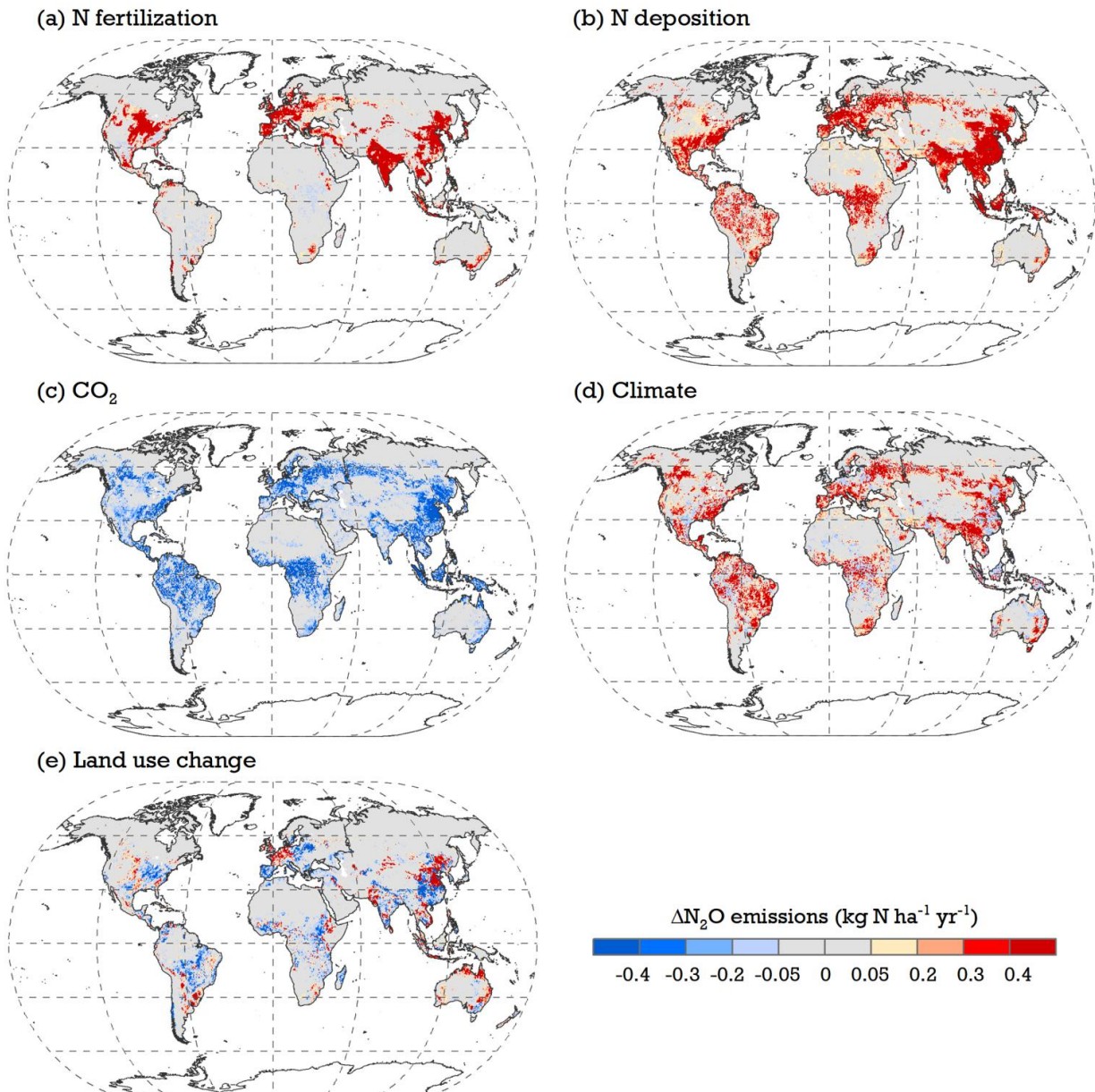

**Figure 9.** Modelled maps of the contributions of (a) N fertilization, (b) N deposition, (c) rising $CO_2$, (d) climate change, and (e) land use change to soil $N_2O$ emissions between 2011–2020. See S1–S6 runs in Table 1 for simulation setups and Eq. 24 for $\Delta N_2O$ calculation.

## 4 Discussion

### 4.1 Model uncertainties at site scale

Incorporating specific nitrification and denitrification processes, together with agricultural management practices, in LPJ-GUESS led to a good agreement between modelled and observed $N_2O$ emissions on cropland sites, despite some overestimations relative to highly fertilized trials. One factor contributing to this overestimation is that some of the processes taking place during crop growth are not well

represented in the model. In previous studies (Ma et al., 2022a, 2023) crop yields simulated by LPJ-GUESS under high N fertilizer inputs were lower than observations, indicating an underestimation of both plant N demand and uptake. Consequently, the excess N remaining in the soil would facilitate higher gaseous loss in the model. This can also explain the significant overestimations on cumulative $N_2O$ emissions in rice cropping systems (Fig. 3), where the simulated growing season was about one month shorter than field experiments since the growth phase between rice sowing and transplanting has not been implemented in LPJ-GUESS. Compared to observations, such a reduction in the simulated growing period was expected to produce lower N uptake and higher $N_2O$ emissions.

We found that the model generally underrepresented $N_2O$ sources in the tropics while simultaneously overestimating annual emissions in temperate and boreal forests across all evaluated natural sites (Fig. S2 in Supporting Information). This discrepancy can be partially attributed to the high levels of soil WFPS simulated for most humid tropical climates with fine- or medium-textured soils (>0.75, not shown), leading to large amounts of $N_2$ gas (instead of $N_2O$) being produced at low oxygen concentrations (Davidson et al., 2000; Pilegaard, 2013). A field-based synthetic analysis estimated that the global mean ratio of $N_2O$ to $(N_2O+N_2)$ on natural soils was 0.125 during the denitrification processes (Scheer et al., 2020). Although our simulated ratios of 0.08–0.10 in tropics and 0.12–0.15 in a typical temperate site were close to this global mean estimate (Fig. S3 in Supporting Information), they may have a potential underestimation of $N_2O$ by 35% in tropical climates and an overestimation by 15% in temperate natural vegetation. This issue could be addressed by adjusting the partitioning scheme between $N_2O$ and $N_2$ in the denitrification processes (see Eq. 21). However, it is currently prevented by the lack of measured $N_2$ data, as accurately determining $N_2$ fluxes from the soil is challenging due to the high background concentration in the atmosphere (Butterbach-Bahl et al., 2013; dos Reis Martins et al., 2022).

LPJ-GUESS simulated the cumulative $N_2O$ emissions in response to variations in soil moisture, temperature, and N fertilizer input reasonably well, but it did not well capture the measured seasonal dynamics, particularly for the representation of peak $N_2O$ daily fluxes (Figs. 4–5). This issue was not unique to our study and has also been reported in other studies using different ecological models (e.g., Gaillard et al., 2018; Huang and Gerber, 2015; Ma et al., 2022b; Val Martin et al., 2023). The poor performance in simulating the variability of daily $N_2O$ fluxes was likely due to the missing or incomplete N transformation processes in the model. For instance, $N_2O$ uptake has often been reported in field studies during low-temperature seasons (Fig. 3), especially in natural ecosystems at high latitudes (Brummell et al., 2014). Recent studies indicated that an $N_2O$ sink could result from efficient $N_2O$ consumption by

anaerobic microsites during the final reduction step of denitrification processes (Hiis et al., 2024; Sihi et al., 2020; Siljanen et al., 2020). However, this underlying mechanism is not well understood, which limits the

610 model's ability to characterize $N_2O$ uptake. In addition, the large differences between simulated and measured $N_2O$ monthly fluxes were seen during the spring when the soil underwent seasonal freezing and thawing (Fig. 3). These $N_2O$ increases associated with freeze-thaw cycles are difficult to simulate because of challenges in parameterizing the transient pulses of nutrient availability in the micro-environment (Zhang et al., 2017b). The impacts of soil freeze events on nutrient release—stimulated by microbial

mortality and the physical breakdown of soil aggregates—have not been represented in either LPJ-GUESS or other process-based models (Tian et al., 2019), despite many field measurements reporting their importance for annual $N_2O$ emissions (Kazmi et al., 2023; Wagner-Riddle et al., 2024).

Similar with other modelling studies, LPJ-GUESS adopted empirical modifiers, $k_{max\_nit}$ (Eq. 6) and $k_{max\_denit}$ (Eq. 13), to simulate maximum nitrification and denitrification rates, in order to account for the limitations

that cannot currently be clarified. These parameters vary to a large degree between models, and this variation likely represents each model's unique structure and the specific N transformation processes it includes (Gabbrielli et al., 2024). In this study, we used prescribed constants for $k_{max\_nit}$ and $k_{max\_denit}$, based on values from the literature (see Sect. 2.2). Assessing the impact of this implementation on seasonal dynamics of $N_2O$ is challenging, as these experiment-dependent parameters are typically unavailable for

each test site. Using fixed values to represent all unclarified nitrification-denitrification situations, such as in our global-uniform parametrization, cannot reflect the high spatial-temporal variability across different sites, which may further affect the evaluated seasonal pattern of $N_2O$ emission at site level.

## 4.2 Global soil $N_2O$ emissions and their drivers

Global simulations by LPJ-GUESS indicated that soil $N_2O$ emissions from natural vegetation were 5.9±0.13

630 Tg N yr$^{-1}$ in the 2010's decade, which was comparable with 6.4 Tg N yr$^{-1}$ reported by the Global Nitrous Oxide Budget (Tian et al., 2024). Saikawa et al. (2014) used the top-down atmospheric inversion method to estimate $N_2O$ emissions from natural soils over the period of 1995–2008, yielding an average emission rate of 7.1 Tg N yr$^{-1}$. Our estimate of 5.8±0.2 Tg N yr$^{-1}$, although lower, remains within these other findings (4.7–8.4 Tg N yr$^{-1}$) for that same 14-year timeframe.

Cropland $N_2O$ emissions were simulated at 3.5±0.25 Tg N yr$^{-1}$ by LPJ-GUESS globally between 2007–2016, which was consistent with the ensemble mean of 3.3 Tg N yr$^{-1}$ by NMIP models over the same period (Tian et al., 2019). Our estimates were also close to those of Xu et al. (2020) and Val Martin et al. (2023), who

reported 3.1–3.2 Tg N yr$^{-1}$ in N$_2$O emissions during 2010–2014 using the DLEM and CLM5 models, respectively. However, these process-based models broadly produced higher N$_2$O emission rates compared with estimates using global inventory approach. For instance, based on the IPCC Tier 1 emission factor (EF), FAOSTAT predicted a 10-year average of 1.7 Tg N yr$^{-1}$ in direct soil N$_2$O emissions from 2011–2020 through the combination of emissions from N fertilizer use, manure application, and crop residues (FAOSTAT, 2024). Similarly, the latest Emissions Database for Global Atmospheric Research (EDGAR v8.0; Crippa et al., 2024) reported direct soil N$_2$O emissions on agricultural lands of 3.0 Tg N yr$^{-1}$ for the same period, including large contributions from livestock excreta on grazing pasture. The discrepancies between our results and EF-based approaches were most likely due to the background "legacy effect"—resulting from soil mineralization and residual N accumulation from previous years—not being accounted for by IPCC guidelines. If considering this "legacy effect" (~1.5 Tg N yr$^{-1}$; Kim et al., 2013), a recent study re-estimated global cropland N$_2$O emissions to be 2.6 Tg N yr$^{-1}$ over 2010–2014, based on an N-rate-dependent EF method (Wang et al., 2020). While our model results are higher, they can thus be seens as comparable with this statistically derived estimate.

Emission factors from IPCC Tier 1 (default values of 0.4% for flooded rice and 1% for other crops; Hergoualc'h et al., 2019) are often used to quantify large-scale soil N$_2$O emissions caused by reactive N inputs. Davidson (2009) implemented an EF of 2.5 % and reported global synthetic fertilizer-induced soil emissions as 2.2 Tg N yr$^{-1}$ in 2005, including both direct and indirect sources. In this study, we only simulated direct N$_2$O emissions from fertilization, which was 2.6 Tg N yr$^{-1}$ in the same year. The higher simulated fertilization effect can be partially attributed to the inclusion of emissions from manure application in the model (S5 in Table 1), whereas Davidson (2009) did not report this agricultural source independently. According to our simulation results in 2005, the ratio of fertilization effect (2.6 Tg N yr$^{-1}$) to total fertilizer inputs (synthetic N and manure; 109 Tg N yr$^{-1}$) was estimated at 2.4% globally, consistent with the findings in Davidson (2009). Notably, both studies imply the default EF of 1% in IPCC Tier 1 might lower the estimation of direct N$_2$O emissions on agricultural soils. Similar with N-applied croplands, recent global meta-analysis has shown that N enrichment (e.g., atmospheric N deposition) significantly increased N$_2$O emissions by 80–101% in natural ecosystems, particularly in temperate and boreal forests (Deng et al., 2020; Shen and Zhu, 2022). We identified N deposition as the most important environmental driver of N$_2$O emissions in our model (S4 in Table 1), contributing to a 126% increase on natural lands during 2011–2020 (Fig. 8). This finding was in line with the results of the meta-data analyses referenced above.

Understanding of the $CO_2$ effect on soil $N_2O$ emissions is still incomplete at the global scale. A quantitative assessment by Van Groenigen et al. (2011) found that rising atmospheric $CO_2$ levels increased $N_2O$ emissions by 25% in upland natural ecosystems. This increase was attributed to enhanced plant fine-root biomass and soil moisture, which favored the carbon availability for denitrifying bacteria under anaerobic conditions. Conversely, a recent meta-analysis argued that elevated $CO_2$ concentrations significantly improved plant N use efficiency by as much as 32%, subsequently decreasing hydrological N loss by 33% and $N_2O$ emissions by 5% in global forests (Cui et al., 2024). It remains unclear which of these two opposing mechanisms might play a more dominant role in field measurements, but the negative $CO_2$ effect on $N_2O$ emissions, at least in our simulations (-87% during 2011–2020 on global natural lands; Fig. 8), can be explained by enhanced vegetation N uptake and reduced soil mineral N surplus under increased $CO_2$ conditions (Fig. S5 in Supporting Information). Studies using other models have shown similar results (e.g., Huang and Gerber, 2015; Tian et al., 2019; Zaehle et al., 2011). However, the situation differs on N-applied agricultural lands, where the high abundance of soil mineral N is typically sufficient for crop uptake and nitrifier-denitrifier use, thus favoring $N_2O$ production even under rising $CO_2$. The low $CO_2$ effect on cropland $N_2O$ emissions compared with natural vegetation emissions in our simulations (-6% during 2011–2020; Fig. 8) are in line with previous modelling findings that reactive N addition (e.g, N fixation and deposition) may diminish the negative influence of elevated $CO_2$ on soil $N_2O$ emissions through mitigating N limitation to both vegetation and soil microbes (Kanter et al., 2016; Xu-Ri et al., 2012).

In this study climate change was simulated to increase global $N_2O$ emissions by 1.3 Tg N yr$^{-1}$ (+24%; Fig. 8) during 2011–2020, which was slightly higher than the mean increase of 1.0 Tg N yr$^{-1}$ by NMIP models (Tian et al., 2019) and below the 33% warming-induced enhancement reported in a meta-analysis (Li et al., 2020). This positive effect in both simulation and observation can be explained by the increased soil temperature, which can (a) speed up the N mineralization process, resulting in N-rich substrates for microbe use, and (b) significantly stimulate the activity and population of denitrifying bacteria, which thrive more in warmer environments than nitrifiers (Pärn et al., 2018). Regionally, a $N_2O$ decline due to climate change was seen in parts of India, Malaysia, and Indonesia (Fig. 9), likely resulting from increased rainfall over the recent two decades (Xu et al., 2020). As discussed in Sect. 4.1, high soil water content linked to ample rainfall can suppress $N_2O$ production in the tropics once soil WFPS exceeds 0.7 (Davidson et al., 2000; Pilegaard, 2013). By contrast, the $N_2O$ decreases in semi-arid climates, such as West Asia, could be attributed to reduced precipitation. In these regions, nitrification is the dominant process

determining N$_2$O production under aerobic conditions and is usually proportional to the simulated soil WFPS (Fig. S3 in Supporting Information; Davidson et al., 2000).

**4.3 Modelling limitations and implications**

Implementing and evaluating soil N$_2$O emissions in models remains challenging due to the short time-scales and high spatial heterogeneity of microbial processes in soils, as well as uncertainties from model input forcings, parametrization, and structure. In this study, we incorporated the influence of major controlling factors—soil moisture and temperature, carbon supply, soil texture and pH, and reactive N availability—to nitrification and denitrification processes in LPJ-GUESS, following empirical findings from previous studies. Although the model can represent the cumulative N$_2$O emissions satisfactorily, the daily magnitude and seasonal variation did not match experimental observations well. This discrepancy probably reflects the differences between highly controlled and/or specific local field conditions and the general protocols adopted for our simulations related to land-use history, N interaction between plants and soils, initial SOM levels, and assumptions made on other N-related fluxes (e.g., NH$_3$ volatilization, N leaching, and partitioning ratio between N$_2$ and N$_2$O). Moreover, some key processes that regulate N$_2$O emissions in field trials—such as the life cycle of nitrifiers and denitrifiers, N$_2$O uptake, heterotrophic nitrification–aerobic denitrification, and freeze-thaw cycles—have not been implemented in the model.

The current LPJ-GUESS version we used only models two soil layers (i.e., 0–50 cm and 50–150 cm; see Smith et al., 2014; Wårlind et al., 2014). Soil organic matter has no defined placement within the soil column, and the N transformation processes implemented in this study are influenced solely by the temperature and moisture of the top layer (see Sect. 2.2). Given the 50 cm thickness, soil water variability during rain events is minimal, which explains the absence of any distinct WFPS peaks in our modelling results (Fig. 5; Fig. S3 in Supporting Information). In reality, however, most N$_2$O-related hydrological processes occur very close to the surface, where soil saturation and drying take place rapidly over short periods. Additionally, soil water content is currently simulated between the wilting point and field capacity (Smith et al., 2014), indicating that fully saturated or completely dry soils cannot be represented in the model. This constraint would consequently impact the simulated WFPS values across differnet soil textures, which, in turn, affects all other N transformation processes relying on WFPS (see Fig. 1). To better account for gaseous N losses from the soil, the improvement of soil hydrological scheme within LPJ-GUESS remains to be taken into account in future model development.

Both modelling and EF-based approaches indicated that managed grasslands were strong $N_2O$ sources, with emissions of 1.5–2.2 Tg N $yr^{-1}$ between 2000–2006, including contributions from livestock excreta deposition, manure use, and N fertilizer application (Dangal et al., 2019; Oenema et al., 2005). In this study, LPJ-GUESS did not account for the effects of N management on pasture, which may have resulted in an underestimation of soil $N_2O$ emissions across global land. However, a long-term gridded dataset for fertilizer and manure application on pasture has recently become available (HaNi; Tian et al., 2022), offering the possibility of incorporating fertilized grasslands into our future model simulations. In addition, concentrations of soil $NH_4^+$ and $NO_3^-$, as essential substrates for nitrifying and denitrifying bacteria, are dominant factors controlling $N_2O$ productions on agricultural lands (Fig. 1). The ratio of $NH_4^+$ to $NO_3^-$, which typically varies with fertilizer types, has been reported to significantly influence soil $N_2O$ emissions in field experiments (e.g., Nelissen et al., 2014; Shcherbak et al., 2014). Globally, Nishina et al. (2017) and Tian et al. (2022) pointed out that the $NH_4^+$:$NO_3^-$ ratio from N fertilizer application gradually increased from 2.0 to 7.0 during 1961–2010 as a result of the increased consumption of urea on croplands. In contrast, LPJ-GUESS assumed this ratio as a constant of 1.0 when agricultural soils received N fertilizer inputs. Using this fixed parameter in our simulations cannot reflect the variability of fertilizer-type-induced $N_2O$ emissions in reality, particularly in highly fertilized regions. Considering the model's sensitivity to N fertilization (Fig. 3) and the contribution of synthetic N fertilizer to global $N_2O$ increases (Fig. 8), a step forward could be to harmonize the existing fertilizer-species-dependent datasets to reduce the simulated uncertainties of cropland $N_2O$ emissions.

In addition to N fertilizer and manure management, conservation agriculture—like reduced tillage, residue retention, and legume cover crops—has for many years been recommended as a promising climate mitigation practice because of its ability to enhance soil carbon sequestration (Poeplau and Don, 2015; Smith et al., 2020). However, much experimental evidence showed that these conservation practices had a potential to offset the $CO_2$ mitigation effect due to the increased $N_2O$ emissions (e.g., Lugato et al., 2018; Mei et al., 2018; Quemada et al., 2020; Yangjin et al., 2021). In this study, we only simulated conventional management because of the limited adoption of conservation agriculture on current global croplands (see Sect. 2.1). Whether such a trade-off between $CO_2$ uptake and $N_2O$ emissions due to conservation practices would also emerge on a large scale—particularly in regions with high N fertilizer applications—needs to be investigated in future work.

**5 Conclusions**

In this study we implemented mechanistic representations of nitrification and denitrification in LPJ-GUESS to account for soil $N_2O$ emissions on global terrestrial ecosystems. The simulated $N_2O$ fluxes from natural soils and croplands were compared against observations ranging from site-level to global scale. Our results showed that the $N_2O$ scheme implemented in the model realistically responded to changes in soil moisture, temperature, and reactive N inputs. It produced cumulative $N_2O$ emission rates comparable with measured data, despite some deviations in seasonal patterns.

In our simulations, global soil $N_2O$ emissions from land ecosystems showed a rapid increase between 1960–2020, rising from 5.6±0.2 Tg N $yr^{-1}$ in the 1960s to 9.9±0.3 Tg N $yr^{-1}$ in the 2010s. While natural vegetation was the predominant $N_2O$ source in the 1960s, its emissions were gradually surpassed by croplands over the study period. During 2011–2020, East Asia emerged as the largest regional source of $N_2O$, with N fertilization (including synthetic fertilizer and manure use) being identified as the major contributor. On average, global $N_2O$ increases due to N fertilization increased from 0.22±0.13 Tg N $yr^{-1}$ in the 1960s to 3.2±0.2 Tg N $yr^{-1}$ in the 2010s. We also found that atmospheric N deposition and climate change have both contributed to the rise of global $N_2O$ emissions, although effects varied significantly among different vegetation types. Conversely, rising $CO_2$ levels was found to reduce simulated $N_2O$ emissions through increased plant N uptake, whereas land-use change had varied spatial effects on emissions depending on how nitrogen was managed after land-cover conversion.

Incorporating key transformations of soil mineral N into LPJ-GUESS offers the opportunity to evaluate total N loss from the soil to the atmosphere, which is essential for accurately quantifying global terrestrial N cycle in response to changing environmental conditions. This representation also facilitates the assessment of climate mitigation potential in global agricultural soils by examining how $CO_2$ uptake and $N_2O$ emissions may interact under various conservation practices.

**Code and data availability**

The LPJ-GUESS model source code is publicly accessible through the Zenodo repository at https://zenodo.org/records/14258279 (Ma et al., 2024). Global historical climate data of CRUJRA are available at https://data.ceda.ac.uk/badc/cru/data/cru_jra/cru_jra_2.4 (Harris et al., 2020; Kobayashi et al., 2015). The land-use data set of HILDA+ can be downloaded at https://doi.pangaea.de/10.1594/PANGAEA.921846 (Winkler et al., 2021). Crop growth distribution from MIRCA2000 can be accessed by https://zenodo.org/records/7422506 (Portmann et al., 2010). The site-

level observations collected from the existing literature, together with large-scale model inputs and outputs, are publicly available through https://zenodo.org/records/14169306 (Ma and Olin, 2024). We use MagicForrest/DGVMTools to analyze the model outputs, with post-processing scripts available at https://github.com/MagicForrest/DGVMTools.

## Author contributions

SO, AA, BS, and JM conceptualized this study. JM, SO, PE, DW, and XR developed the model code. JM, AA, and SO designed the model protocol runs. JM carried out the formal analysis and visualization. SO assisted with field data collection and parameter tuning for model evaluation. PA and MW processed the model input forcing globally. BS, DW, AA, and PA provided constructive suggestions for the discussion on model limitations. JM wrote the original draft, with further editing from all authors.

## Competing interests

The authors declare that they have no conflict of interest.

## Acknowledgements

This research has been supported by the European Union's Horizon Europe research and innovation programme (EYE-CLIMA) under Grant Agreement No. 101081395, and the Swedish Research Council FORMAS (Grant No. 211-2009-1682). This study is a contribution to the Strategic Research Areas Biodiversity and Ecosystem Services in a Changing Climate (BECC) and Modelling the Regional and Global Earth System (MERGE) funded by the Swedish Government.

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
