# Peer review of "Soil nitrous oxide emissions from global land ecosystems and their drivers within the LPJ-GUESS model (v4.1)"

_Geoscientific Model Development, 2024_

## Author Comment (AC1)

**Response to anonymous referee #1** (Comments in black, Answers in blue, Suggested revisions in green)

**General comments:**

*The paper under my reviewing addresses an important topic and contributes to improving the understanding of soil $N_2O$ emissions from global land ecosystems. Overall, the authors have done a great job. They integrated soil nitrification-denitrification processes into the DGVM LPJ-GUESS model, clearly explaining and documenting the added mathematical formulations and parameters, although some complex processes were simplified during the development due to the "large-scale" model structure. Additionally, they examined the model's performance using a valuable site-level data set and quantified the environmental factors driving changes in global $N_2O$ emissions among terrestrial ecosystems through model-based scenario analyses. I think this topic is interesting and of great relevance to GMD. The manuscript is in general well-written and easy to follow. However, a few pieces of information are missing, and some sections require minor adjustments before it can be accepted for publication. Below, I provide specific comments, which I hope will help further improve the manuscript.*

We thank the reviewer for the expressed interest in our manuscript. In the revisions to the manuscript we will be addressing the raised questions as described below.

**Specific comments:**

*The introduction is well stated. The description is sufficient to understand what has been developed in previous DGVM studies in terms of $N_2O$-related processes. However, the organization in some parts can be improved. LN65-66: 0.2-1.8% of what? Is it the proportion of N fertilization lost as $N_2O$?*

Yes, '0.2-1.8%' is the percent of the N inputs, including the N applied as synthetic and organic fertilizer, crop residues, and the N deposited by grazing animals etc. To make it clear, "0.2-1.8%" will be revised to "0.2-1.8% of the N applied the soil" in the manuscript as suggested.

*LN97-100: What specific cropland management practices are included in your model? How are these practices set up in the global simulations? Since conservation agriculture can significantly influence $N_2O$ fluxes in the fields, even though most practices mitigate soil $CO_2$ emissions by enhancing soil C storage. It is better the specify the management setups somewhere in the methodology section.*

At this moment, the cropland management practices implemented in LPJ-GUESS include tillage, crop residue retention, industrial N fertilizer and manure application, and cover cropping (both leguminous and non-leguminous). Given the widespread adoption of conventional practices in global agricultural soils, our large-scale simulations in this study assume that all croplands are managed with tillage and without cover cropping systems, with 25% of aboveground crop residue retained in the fields after harvest. Additionally, all crops receive time-dynamic N fertilizer and manure inputs over the years, with the timing of application varying by crop type.

We clarified the information on management practice setups in the first paragraph in Sect. 2.1 in the original manuscript: "Agricultural practices—such as tillage intensity, N mineral fertilizer and manure application, crop residue removal, and leguminous and non-leguminous cover crops—are also included (Ma et al., 2023; Olin et al., 2015b; Pugh et al., 2015). For large-scale application, to reflect the widespread adoption of conventional practices on current global agriculture (Porwollik et al., 2019), the model assumes that all croplands are under tillage management without cover cropping systems, and that 25% of aboveground crop residue is retained in the fields after harvest. Industrial N fertilizer is added to soils at three different stages of crop growth, with application rates varying by CFT. In contrast, all manure is applied as a single input at crop sowing to reflect the time required for manure N to become available for plant uptake in real-world practices (Olin et al., 2015b)."

*Fig.1: In nitrification processes, where is the heterotrophic pathway? How do you consider this in the model depending on DON? This is an important process in acidic soils or environments where autotrophic nitrification is less dominant. It should be mentioned in the model description.*

We agree with the reviewer that heterotrophic nitrification plays an important role in acidic soil environments. However, in this study, we consider only autotrophic nitrification, as it is the dominant pathway in most natural and agricultural soils (Chapin III et al., 2011). Additionally, modeling the heterotrophic pathway is more challenging because it requires estimating dissolved organic nitrogen (DON) as the main substrate for heterotrophic nitrifiers, and the relevant DON processes have not yet been implemented in LPJ-GUESS.

We will modify the information of heterotrophic nitrification in Sect. 2.2.2 in the revised manuscript. Suggested revision: "Autotrophic nitrification and heterotrophic nitrification are two distinct biological processes involved in the N transformation in soil ecosystems. We focused solely on representing autotrophic nitrification, which is the dominant process in most natural and agricultural soils (Chapin III et al., 2011). The heterotrophic pathway is also more challenging to model as it requires estimation of dissolved organic nitrogen as the main substrate for the responsible nitrifying bacteria."

*LN191-195: Maybe the 38℃ is needed further discussion to represent the entire nitrification process, as NOB bacteria significantly favor higher temperatures compared to AOB and AOA.*

True, as the reviewer mentioned, NOB bacteria significantly prefer higher temperatures than AOB and AOA. However, LPJ-GUESS currently does not simulate the growth and mortality of soil microbes, making it challenging to independently represent the temperature effects on different bacterial groups in the model.

We clarified this limitation in Sect. 2.2.2 in our original submission (Lines 182-184): "Due to the current limitation in the model's ability to simulate the growth and mortality of soil microbes, we integrate these two oxidation steps into one single process—i.e., $NH_4^+$ is oxidized to $NO_3^-$ directly—to collectively represent nitrification in LPJ-GUESS (see Fig.1)."

In addition, to clarify, we will provide an explanation in the revised manuscript for why 38℃ was chosen in Eq. 7. Suggested revision: "Soil temperature plays a crucial role in regulating microbial activities. For nitrite-oxidizing bacteria, 37–39℃ is found to be optimal for substrate oxidation (Taylor et al., 2019) and for ammonia-oxidizing bacteria and archaea the optimal soil temperature can range from 31–42℃ (Ouyang et al., 2017). In the model, the maximum nitrification rate is thus assumed to occur at 38℃, as the average optimal temperature for these three groups of nitrifiers."

*LN420-424: It would be valuable to compare crop yields (or N use efficiency). For instance, in the high N fertilization scenario, was the modelled yield lower than the observed yield? If so, this could suggest that less N was taken up, resulting in less N removal from the system and leaving more N to be emitted as N₂O.*

Good point. We did not compare the yield in this study, but did it in our previous studies (see Ma et al., 2022a; Ma et al., 2023). The model generally generates lower yields than observations at the high fertilizer levels, likely resulting in excess reactive N remaining in the soils and higher N₂O loss.

We have discussed this in the first paragraph in Sect. 4.1 in the original manuscript (Lines 576-583): "In previous studies (Ma et al., 2022a, 2023) crop yields simulated by LPJ-GUESS under high N fertilizer inputs were lower than observations, indicating an underestimation of both plant N demand and uptake. Consequently, the excess N remaining in the soil would facilitate higher gaseous loss in the model. This can also explain the significant overestimations on cumulative N₂O emissions in rice cropping systems (Fig. 3), where the simulated growing season was about one month shorter than field experiments since the growth phase between rice sowing and transplanting has not been implemented in LPJ-GUESS. Compared to observations, such a reduction in the simulated growing period was expected to produce lower N uptake and higher N₂O emissions."

*LN450-453: I would suggest to add the observed WFPS to Fig. S3 to clearly compare the mismatch between the simulation and observation, if the reported data are available.*

We understand the reviewer's concerns regarding the comparison of modeled and observed soil moisture at the site level. Unfortunately, the reported WFPS at the three field sites in Fig. S3 is not consistently available throughout the experimental periods. For instance, at Tapajós National Forest in Brazil (Fig. S3a; Davidson et al., 2008), only volumetric soil water content at a depth of 2 m was reported, which cannot be directly compared with the simulated WFPS at 0.5 m. At the temperate forest in Japan (Fig. S3b; Morishita et al., 2007), no observed soil moisture data were available. For the semi-arid grassland in China (Fig. S3c; Du et al., 2006), only a few monthly (seasonal) WFPS values were reported for 1998 (2001), despite the total experimental period lasting five years. Accordingly, it is challenging to add the observed WFPS to Fig. S3.

*LN482-484 and Fig.5d: It could be interesting to identify which thin-dashed blue WFPS corresponds to each N treatment. Does the N250 scenario, with the highest N, use the most water in the soil?*

Yes, in the model, crops with the highest N250 input consume the most soil water, resulting in the lowest remaining WFPS in the soil. As the reviewer suggested, we will clearly label the thin-dashed blue WFPS line for each N treatment in Fig. 5d, which will be updated accordingly in the revised manuscript as below:

[Figure]

*LN545-547: It seems to find an explanation for the reduced $N_2O$ emissions under elevated $CO_2$ concentrations.*

In LPJ-GUESS, rising $CO_2$ levels in the atmosphere reduce the simulated $N_2O$ emissions primarily due to increased plant N uptake (see Fig. S5). The reduced N remaining in the soil leads to lower gaseous N emissions to the atmosphere.

We gave an in-depth discussion in the fourth paragraph in Sect. 4.2 in our original manuscript: "It remains unclear which of these two opposing mechanisms might play a more dominant role in field measurements, but the negative $CO_2$ effect on $N_2O$ emissions, at least in our simulations (-87% during 2011-2020 on global natural lands; Fig. 8), can be explained by enhanced vegetation N uptake and reduced soil mineral N surplus under increased $CO_2$ conditions (Fig. S5 in Supporting Information). Studies using other models have shown similar results (e.g., Huang and Gerber, 2015; Tian et al., 2019; Zaehle et al., 2011)."

*LN651-660: For large-scale applications, fertilizer type-usually simplified as the ratio of NH4 to NO3 in the model-is one of the factors contributing to the simulated uncertainty in N2O emissions. Does LPJ-GUESS actually account for this when performing the S5 runs ("Const_Nfert")?*

We agree with the reviewer's opinion that fertilizer type significantly affects the uncertainty in simulated $N_2O$ fluxes, as soil NH4 and NO3 are essential substrates for nitrifying and denitrifying bacteria that regulate $N_2O$ production. Currently, LPJ-GUESS does not distinguish between different fertilizer types and assume a fixed 1:1 ratio of NH4 to NO3 for all N fertilizer inputs.

We will bring this limitation to Sect. 4.3 for discussion: "In addition, concentrations of soil $NH_4^+$ and $NO_3^-$, as essential substrates for nitrifying and denitrifying bacteria, are dominant factors controlling $N_2O$ productions on agricultural lands (Fig. 1). The ratio of $NH_4^+$ to $NO_3^-$, which typically varies with fertilizer types, has been reported to significantly influence soil $N_2O$ emissions in field experiments (e.g., Nelissen et al., 2014; Shcherbak et al., 2014). Globally, Nishina et al. (2017) and Tian et al. (2022) pointed out that the $NH_4^+:NO_3^-$ ratio from N fertilizer application gradually increased from 2.0 to 7.0 during 1961–2010 as a result of the increased consumption of urea on croplands. In contrast, LPJ-GUESS assumed this ratio as a constant of 1.0 when agricultural soils received N fertilizer inputs. Using this fixed parameter in our simulations cannot reflect the variability of fertilizer-type-induced $N_2O$ emissions in reality, particularly in highly fertilized regions."

**References**

[revised manuscript text omitted]

---

## Author Comment (AC2)

**Response to anonymous referee #2** (Comments in black, Answers in blue, Suggested revisions in green)

**General comments:**

*Ma et al. describe a new parametrization for soil mineral N transformation in the widely used DGVM LPJ-GUESS model. They mainly take NH₃ volatilization, nitrification, and denitrification processes into account, with well explaining the mechanistic parametrization of N dynamics. They also provide a global evaluation of the model development in terms of simulated N₂O fluxes among land ecosystems. Overall this is a well written paper with a good structure in simulation experiment design and assessment results. Although modelling soil N₂O emissions is a challenging work with large uncertainties, authors do a good job by giving a solid discussion on model limitations of N transformation which are well referenced. I think this is an exciting and timely work that greatly advances existing modeling tool and will benefit to biogeochemical research community, it can be accepted by GMD for publication after minor adjustments.*

We thank the reviewer for the expressed interest in our manuscript. In the revisions to the manuscript we will be addressing the raised questions as described below.

**Specific comments:**

*The new model development section is clear and thorough; however, the model simulation protocol needs to be further clarified. For instance, in sections 2.3.1 and 2.3.2, are the forcing data (e.g., climate and N deposition) the same for the site-specific and global runs? Additionally, how does LPJG estimate the dynamics of soil pH during N transformation processes? A brief description of the model's representation of soil physical properties should also be included in section 2.3.*

Thanks for this comment. In this study, the climate forcing and N deposition inputs are identical between the site-specific and global runs. We have clarified this in our original manuscript (Lines 334-337): "Due to the lack of weather and N deposition data for most study sites, an observation-based gridded climate dataset, CRUJRA v2.4 (Harris et al., 2020; Kobayashi et al., 2015), and an atmospheric N deposition dataset simulated by CCMI ($NH_x$-N and $NO_y$-N; Tian et al., 2018), were used as inputs to drive LPJ-GUESS, selecting the representative grid cell (0.5°×0.5°) for each experimental site." Also in Lines 352-357: "For global-scale applications, climate variables—daily temperature (minimum, mean, and maximum), precipitation, solar radiation, specific humidity, and wind speed from CRUJRA v2.4 dataset—were used for driving model simulations, ranging from 1901–2020 at a resolution of 0.5°×0.5° (Harris et al., 2020; Kobayashi et al., 2015). Historical annual atmospheric $CO_2$ concentration and monthly N deposition rates over the same period were derived from Meinshausen et al. (2020) and Tian et al. (2018), respectively."

Soil physical characteristics, such as texture and pH, are provided as external inputs to LPJ-GUESS for hydraulic property calculation. All these physical characteristics are treated as constant throughout the experimental periods. To make it clear, we will add a description in Sect. 2.3.1. Suggested revision: "Additionally, to estimate soil hydraulic properties and evaluate our developed N transformation processes, soil physical characteristics—such as soil pH and texture (i.e., content of sand, silt, and clay)—were collected from the literature and kept constant during the simulation period."

*The site-level evaluation is convincing, but the global-scale results require further discussion. Based on Figures 6 and 7, the simulated N2O emissions from pastures are not entirely convincing. It would be expected to see higher N2O fluxes from pastures starting in the 1980s onward, given the increased use of manure and fertilizers during this period. Additionally, how does the model account for manure deposition from livestock waste on grazing pastures? This aspect should be clarified to better understand the model's representation of pasture emissions.*

We agree with the reviewer's comment that the simulated N₂O fluxes from pastures since the 1980s in our study are generally lower than other estimates. This underestimate can be attributed to two main reasons: (a) the current version of LPJ-GUESS used in this study is not equipped with N fertilization on pastures, the relevant work is still under development; and (b) before our N₂O development work was completed, we had not found a suitable global N application data set for pastures (including chemical

fertilizer and manure use). Overall, the exclusion of fertilized pastures is the primary driver for the low $N_2O$ fluxes simulated on global pastures in this study. To avoid confusion, we have clarified this limitation throughout the original manuscript. For instance, in Abstract (Line 33): "Our estimates only account for the direct soil $N_2O$ emissions, excluding those from fertilized pasture". Also in Sect. 2.3.2 (Lines 366-367): "At present, pasture ecosystems represented in LPJ-GUESS do not receive any N fertilizer inputs". In addition, we have compared our results with other findings and gave an in-depth discussion in Sect. 4.3 (Lines 727-733): "Both modelling and EF-based approaches indicated that managed grasslands were strong $N_2O$ sources, with emissions of 1.5-2.2 Tg N $yr^{-1}$ between 2000-2006, including contributions from livestock excreta deposition, manure use, and N fertilizer application (Dangal et al., 2019; Oenema et al., 2005). In this study, LPJ-GUESS did not account for the effects of N management on pasture, which may have resulted in an underestimation of soil $N_2O$ emissions across global land. However, a long-term gridded dataset for fertilizer and manure application on pasture has recently become available (HaNi; Tian et al., 2022), offering the possibility of incorporating fertilized grasslands into our future model simulations."

In the default setup of LPJ-GUESS, 75% of the harvested N from aboveground biomass is assumed to return to the soil annually, in order to represent internal manure deposition from livestock on grazing pastures. To make it clear, we will bring this description to Sect. 2.1. Suggested revision: "To account for internal manure deposition from livestock on pastures, the model assumes that 75% of the nitrogen from the harvested biomass is returned to the soil (Lindeskog et al., 2013)."

*Line78: "… (DGVMs)", a citation of DGVM would be appreciated.*

A citation will be added as the reviewer suggested. In the revised manuscript, we will change the sentence accordingly: "The implementation of $N_2O$-related processes into global biosphere models, such as Dynamic Global Vegetation Models (DGVMs; Cramer et al., 2001) only began in the early years of this century."

*Line 119 "…of a proportion of the aboveground biomass", What is the proportion used in the model? It appears to be a dynamic variable that varies depending on the study region, but what is the default value applied in your global simulation? This information is important for understanding the baseline assumptions and how they influence the results.*

In the default global configuration, half of the aboveground biomass is harvested every year to account for the grazing effects. As the reviewer suggested, this sentence will be corrected in the revised manuscript: "Pasture ecosystems are modelled as a competition between $C_3$ and $C_4$ grasses, with 50% of the aboveground biomass removed annually to represent grazing effects."

*Line347: Could you provide specific details on how the soil physical characteristics are set up at the study sites? Are these characteristics treated as constant throughout the experimental periods, or are they modeled as time-dynamic variables?*

Thanks for this comment. Please refer to our previous responses above.

*Figure 4c: Why is there such a large mismatch between the simulation and observations in April?*

Most field measurements have reported transit pulses of $N_2O$ during spring in high-latitude natural land ecosystems, when soils undergo seasonal freezing and thawing. This is largely due to enhanced nutrient release during soil freeze events, stimulated by microbial mortality and the physical breakdown of soil aggregates. However, these processes associated with freeze-thaw cycles have not yet been implemented in LPJ-GUESS or other models. This explains the lower simulated $N_2O$ fluxes than field observations in April shown in Fig. 4c.

We have included this discussion in the third paragraph of Sect. 4.1 in our original manuscript: "In addition, the large differences between simulated and measured $N_2O$ monthly fluxes were seen during the spring when the soil underwent seasonal freezing and thawing (Fig. 3). These $N_2O$ increases associated with freeze-thaw cycles are difficult to simulate because of challenges in parameterizing the transient pulses of nutrient availability in the micro-environment (Zhang et al., 2017b). The impacts of soil freeze events on nutrient release—stimulated by microbial mortality and the physical breakdown of soil aggregates—have not been represented in either LPJ-GUESS or other process-based models (Tian et al., 2019), despite many field measurements reporting their importance for annual $N_2O$ emissions (Kazmi et al., 2023; Wagner-Riddle et al., 2024)."

*Sect. 3.1.2: What is the soil texture at the study sites? In most Dynamic Global Vegetation Models, the simulated WFPS is highly sensitive to soil texture, with very fine soils typically exhibiting higher WFPS. An accurate representation of hydrology in the model would significantly improve the simulation of all N₂O-related processes that depend on WFPS (see your Figure 1). Since different soil texture classes undoubtedly affect WFPS, a brief explanation of how WFPS is associated with soil texture in your model would be greatly appreciated.*

Soil texture information for all evaluated sites, collected from the literature, is publicly available through the Zenodo repository (https://doi.org/10.5281/zenodo.14169306). In the revised manuscript, we will also include soil texture information when comparing the simulated and observed WFPS at the study sites. For example, the addition in Sect. 3.1.2: "Soil moisture was identified as the dominant factor controlling the seasonal dynamics of $N_2O$ fluxes at a semi-arid grassland with a sandy loam soil texture, where observed WFPS ranged from 0.01-0.48 between June and August (Du et al., 2006). While the model effectively represented this $N_2O$ rise due to increasing WFPS, it overestimated total emissions by ~95% over the summer season (Fig. 4h). This overestimation primarily resulted from the model simulating a higher WFPS value of 0.37 for this sand-dominant soil, compared with the observed average of 0.20 in these three months". Also, in Sect. 3.1.3: "At this site with silt-clay textured soil, the simulated seasonal trends of soil WFPS and temperature broadly aligned with the observed variations (Pearson correlation coefficients of 0.42-0.57, $p < 0.05$ for both variables), despite the modelled WFPS values being consistently higher than the observed ones."

We agree with the reviewer's perspective that accurately representing WFPS based on soil texture can improve the model's performance in simulating $N_2O$ fluxes. Therefore, an in-depth discussion of soil moisture limitations on the simulated $N_2O$ emissions was provided in our original manuscript (Lines 717-729): "The current LPJ-GUESS version we used only models two soil layers (i.e., 0-50 cm and 50-150 cm; see Smith et al., 2014; Wårlind et al., 2014). Soil organic matter has no defined placement within the soil column, and the N transformation processes implemented in this study are influenced solely by the temperature and moisture of the top layer (see Sect. 2.2). Given the 50 cm thickness, soil water variability during rain events is minimal, which explains the absence of any distinct WFPS peaks in our modelling results (Fig. 5; Fig. S3 in Supporting Information). In reality, however, most $N_2O$-related hydrological processes occur very close to the surface, where soil saturation and drying take place rapidly over short periods. Additionally, soil water content is currently simulated between the wilting point and field capacity (Smith et al., 2014), indicating that fully saturated or completely dry soils cannot be represented in the model. This constraint would consequently impact the simulated WFPS values across different soil textures, which, in turn, affects all other N transformation processes relying on WFPS (see Fig. 1). To better account for gaseous N losses from the soil, the improvement of soil hydrological scheme within LPJ-GUESS remains to be taken into account in future model development."

*Figure 7: Here again, the low N₂O emission on global grazing pasture makes non-sense to me due to the livestock waste inputs and N fertilizer use in the recent decades.*

Thanks for this comment. Please refer to our previous responses above.

*Line745-752: How do N₂O emissions in the model respond to different agricultural management practices, such as fertilizer application, tillage, or crop rotation? How closely do the simulated results align with real-world observations? Additionally, are these management practices explicitly included in your global simulations? If not, I think your discussion is not strongly supported.*

Cropland management practices implemented in LPJ-GUESS include tillage, crop residue retention, industrial N fertilizer and manure application, and cover cropping (both leguminous and non-leguminous). Given the widespread adoption of conventional practices in global agricultural soils, our large-scale simulations in this study assume that all croplands are managed with tillage and without cover cropping systems, with 25% of aboveground crop residue retained in the fields after harvest. Additionally, all crops receive time-dynamic N fertilizer and manure inputs over the years, with the timing of application varying by crop type. We have clarified these management practice setups in the first paragraph of Sect. 2.1 in the original manuscript already (Lines 125-133).

Unfortunately, we do not have any results on $N_2O$ emissions in response to different agricultural management practices on a global scale at present, which is outside the scope of our current study. However, as a case study, we primarily investigated the influence of the above-mentioned practices on cropland $N_2O$ fluxes over Europe during the historical period (see the Table and Figure below; please note that the results presented below have not been published yet). Preliminary simulation results suggest that, compared with the standard management (STD), using cover cropping (especially for the legume cover crops $CC_L$) and 100% of residue retention (RR) practices would significantly enhance cropland $N_2O$ emissions, while no-tillage system (NT) has the potential to mitigate gaseous N emissions to the atmosphere. Although these magnitudes of $N_2O$ fluxes under different management practices by LPJ-GUESS have not been evaluated or compared against other existing findings, the model does realistically reflect the responses of $N_2O$ emissions to conservation agriculture to some extent. Therefore, we believe it is reasonable to focus our future work on the trade-off between $CO_2$ uptake and $N_2O$ emissions resulting from conservation practices on a large scale (see Lines 745-752 in the original manuscript).

[Figure]

**Simulation setups representing various cropland management practices over Europe**

|  | $CC_L$ | $CC_{NL}$ | RR | NT | STD |
|---|---|---|---|---|---|
| Legume cover crop | Yes | No | No | No | No |
| Non-legume cover crop | No | Yes | No | No | No |
| Residue retention | 25% | 25% | 100% | 25% | 25% |
| Manure application | Yes | Yes | Yes | Yes | Yes |
| Mineral N fertilizer | Yes | Yes | Yes | Yes | Yes |
| Tillage | Yes | Yes | Yes | No | Yes |

**References**

Cramer, W., Bondeau, A., Woodward, F. I., Prentice, I. C., Betts, R. A., Brovkin, V., Cox, P. M., Fisher, V., Foley, J. A., Friend, A. D., Kucharik, C., Lomas, M. R., Ramankutty, N., Sitch, S., Smith, B., White, A. and Young-Molling, C.: Global response of terrestrial ecosystem structure and function to CO2 and climate change: Results from six dynamic global vegetation models, Glob. Chang. Biol., 7(4), 357–373, doi:10.1046/j.1365-2486.2001.00383.x, 2001.

[revised manuscript text omitted]